# RANKNOVO: A UNIVERSAL RERANKING APPROACH FOR ROBUST DE NOVO PEPTIDE SEQUENCING

## ABSTRACT

De novo peptide sequencing is a critical task in proteomics research. However, the performance of current deep learning-based methods is limited by the inherent complexity of mass spectrometry data and the heterogeneous distribution of noise signals, leading to data-specific biases. We present RankNovo, the first deep reranking framework that enhances de novo peptide sequencing by leveraging the complementary strengths of multiple sequencing models. RankNovo employs a list-wise reranking approach, modeling candidate peptides as multiple sequence alignments and utilizing axial attention to extract informative features across candidates. Additionally, we introduce two new metrics, PMD (Peptide Mass Deviation) and RMD (Residual Mass Deviation), which offer delicate supervision by quantifying mass differences between peptides at both the sequence and residue levels. Extensive experiments demonstrate that RankNovo not only surpasses its individual base models, which are used to generate training candidates for reranking pre-training, but also sets a new state-of-the-art de novo sequencing benchmarks. Moreover, RankNovo exhibits strong zero-shot generalization to unseen models—those whose generations were not exposed during training, highlighting its robustness and potential as a universal reranking framework for peptide sequencing. Our work presents a novel reranking strategy that fundamentally challenges existing single-model paradigms and advances the frontier of accurate de novo peptide sequencing. Our source code is provided at an anonymous link [1].

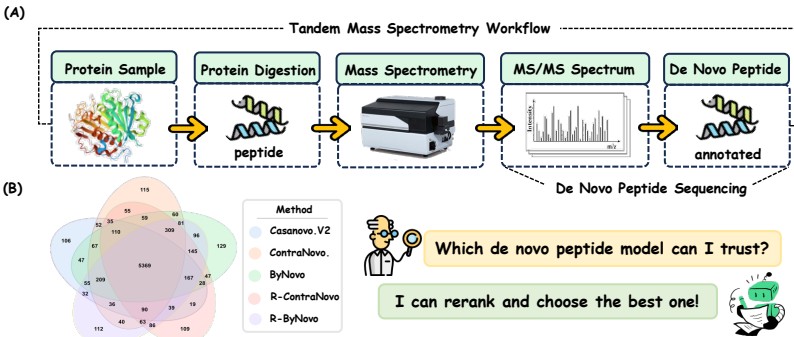

Figure 1: (A) **De Novo Peptide Sequencing Workflow Using Tandem Mass Spectrometry**: Our objective is to predict peptide sequences from MS/MS spectra, as illustrated in the final two steps. (B) **Motivation for RankNovo**: Current de novo peptide sequencing models exhibit data preference in their peptide predictions. Our proposed RankNovo improves overall prediction accuracy by ensembling and reranking the outputs of these models to identify the optimal sequence.

## 1 INTRODUCTION

Identifying proteins is a critical task in proteomics, with mass spectrometry-based shotgun proteomics being widely regarded as the predominant technique for this purpose (Aebersold & Mann,

---

[1]https://anonymous.4open.science/r/RankNovo-F2FB

2003). As shown in Figure 1, this process begins with the enzymatic digestion of proteins into smaller peptide fragments, which are then analyzed using tandem mass spectrometry (MS/MS) to generate spectra (Nesvizhskii et al., 2003). These spectra are subsequently interpreted to infer peptide sequences, enabling precise identification and characterization of proteins. This foundational approach is pivotal for advancing research in proteomics (Aebersold & Mann, 2003).

Proteomics utilizes two primary methodologies for peptide sequence identification: database searching (Ma et al., 2003; Chen et al., 2020) and de novo sequencing (Dančík et al., 1999). In database searching, experimental spectra are matched against pre-existing entries in protein databases to identify the most likely sequences. Although effective for identifying known peptides, this approach is inherently constrained by the completeness of the database, posing challenges when encountering novel or uncharacterized sequences (Karunratanakul et al., 2019; Hettich et al., 2013). On the other hand, de novo sequencing leverages the intrinsic patterns of tandem mass spectra to directly infer peptide sequences without requiring a reference database. This enables the discovery of novel peptides, overcoming the limitations inherent in database-dependent approaches. Consequently, de novo sequencing has emerged as a critical technique for peptide identification, significantly advancing the scope of proteomic analysis (Ng et al., 2023).

Over the past two decades, de novo sequencing has made substantial progress, evolving from graph-theoretic and dynamic programming-based methods to more sophisticated approaches driven by deep learning (Ma et al., 2003; LeCun et al., 2015). DeepNovo (Tran et al., 2017) was the first to apply deep learning to de novo sequencing, which inspires a series of subsequent models(Zhou et al., 2017; Karunratanakul et al., 2019; Yang et al., 2019; Liu et al., 2023). More recently, Transformer architectures are introduced to model de novo sequencing as a machine translation task (Yilmaz et al., 2022; Mao et al., 2023; Eloff et al., 2023; Yang et al., 2024; Xia et al., 2024). Building upon this foundation, ContraNovo (Jin et al., 2024) further advanced the field by incorporating multimodal alignment strategies, achieving state-of-the-art performance.

Despite recent advancements in de novo peptide sequencing, these methods still exhibit notable accuracy limitations when compared to traditional database search approaches (Muth et al., 2018). The primary challenge stems from the inherent complexity of mass spectrometry data, which consists of a mixture of heterogeneous distributions. This complexity is driven by variations in experimental conditions, such as differences in instrumentation, protocols, and target protein species, each of which introduces distinct noise patterns into the acquired spectra (Zubarev & Mann, 2007; Chang et al., 2016). As shown in Fig. 1(B), no model is exempt from issues of generalization and preferential bias, as evidenced by the presence of unique correct predictions from models that otherwise exhibit weaker overall performance. This observation motivates a rethinking of de novo peptide sequencing as a reranking task, where a trained meta-model selects the optimal prediction from a collection of outputs generated by multiple de novo models.

In this paper, we introduce RankNovo, a novel deep reranking framework designed to address the preferential bias challenges inherent in peptide sequencing. In such a complex task, peptide candidates generated for the same spectra often exhibit only minor mass differences. To effectively differentiate between these closely related candidates, RankNovo employs a list-wise reranking approach, processing and reranking all candidates in a single forward pass. This strategy enables the model to incorporate information across candidates, facilitating more precise discrimination between similar sequences. This approach stands in contrast to traditional pair-wise comparison frameworks commonly used in Natural Language Processing tasks (Ouyang et al., 2022; Jiang et al., 2023). To implement this reranking strategy, RankNovo formulates peptide candidates as a Multiple Sequence Alignment (MSA) (Jumper et al., 2021; Rao et al., 2021; Abramson et al., 2024) and applies axial attention to extract sequential features. In particular, column-wise attention plays a crucial role in enabling the flow of information and intricate comparisons between candidates (Huang et al., 2019; Ho et al., 2019; Wang et al., 2020). Additionally, spectrum features are extracted using a Transformer encoder and integrated into the peptide track via a cross-attention mechanism.

Moreover, the key concentration on amino acid masses in de novo peptide sequencing (Jin et al., 2024) inspires us to propose two novel metrics, PMD (Peptide Mass Deviation) and RMD (Residual Mass Deviation), as a more nuanced replacement of typical reranking losses such as binary classification loss. The two metrics quantitatively evaluate the mass difference between peptides at both the peptide and residue levels to provide more accurate supervision scores for RankNovo.

Experimental results show that RankNovo achieves state-of-the-art performance on de novo sequencing benchmarks, outperforming each of its component base models, including the current SOTA model, ContraNovo. We also conducted detailed analytical and ablation studies to verify the robustness of the model. Furthermore, we demonstrate that RankNovo, when trained on specific base models, can be effectively applied in a zero-shot setting to peptide predictions from unseen sequencing models, highlighting its strong transferability and its ability to capture deep knowledge for assessing peptide-spectrum matching performance.

The contributions of this paper can be summarized as follows: (1) We introduce the first deep learning-based reranking framework for peptide de novo sequencing, designed to bridge the gap between existing methods, thereby unleashing their complementary potentials. (2) We propose RankNovo, a list-wise reranking framework that models candidates as multiple sequence alignments (MSA) and uses axial attention to extract informative features. (3) We further introduce two novel metrics, PMD and RMD, for accurate measurement of mass differences between peptides, providing precise supervised signals for reranking models. (4) Extensive experiments demonstrate that RankNovo not only surpasses each of its individual ensemble components but also generalizes effectively to unseen models in a zero-shot setting, highlighting its robustness and adaptability.

## 2 RELATED WORK

### 2.1 DE NOVO PEPTIDE SEQUENCING

De novo sequencing holds the potential to overcome the limitations inherent in traditional database search-based methods, thereby making the enhancement of its accuracy a critical objective (Frank & Pevzner, 2005). Early de novo sequencing techniques primarily utilized dynamic programming algorithms along with various scoring functions to evaluate candidate peptide sequences (Ma et al., 2003; Chi et al., 2010; Ma, 2015). Recent advancements have incorporated neural networks into peptide de novo sequencing. These modern methods leverage the powerful generalization capabilities of deep learning (LeCun et al., 2015; Tran et al., 2017; Yang et al., 2019; Qiao et al., 2021; Liu et al., 2023), thereby addressing many computational issues encountered by traditional approaches. Recently, Transformer is applied to de novo peptide sequencing, enabling the direct prediction of peptide sequence (Yilmaz et al., 2022; Mao et al., 2023; Eloff et al., 2023; Yang et al., 2024; Xia et al., 2024). Building on this foundation, ContraNovo (Jin et al., 2024) further enhances the accuracy of de novo sequencing by introducing a contrastive learning training strategy.

Despite these advancements, current methods still face inherent limitations due to the complexity of spectra. In this study, we aim to address these limitations by initially obtaining candidate peptides for each spectrum based on the predictions of several state-of-the-art de novo models. We then develop an effective reranking model to select the best matching candidate, thereby enhancing the overall capability of the de novo sequencing algorithm.

### 2.2 CANDIDATE RERANKING

In the reranking task, methods are typically categorized into three types: point-wise, pair-wise, and list-wise (Zhuang et al., 2023). The point-wise method independently evaluates the relevance of a single query-candidate pair (Nogueira et al., 2019; 2020). The pair-wise method assesses the relative relevance between two candidate pairs for a given query (Burges et al., 2005; Burges, 2010; Ouyang et al., 2022; Jiang et al., 2023). The list-wise method considers the relevance of all candidate pairs for each query collectively, utilizing all candidate features, which enhances performance potential (Han et al., 2020; Gao et al., 2021; Ren et al., 2021). Building on these methods, our study focuses on ranking peptide candidates from selected weak models to identify the best match. We rerank the candidate peptides from all base models using a list-wise strategy, evaluating each one in a single forward pass. This is powered by a axial-attention-based peptide encoder, adept at discerning subtle nuances among candidates, thereby achieving precise differentiation.

### 2.3 AXIAL ATTENTION

Axial attention (Huang et al., 2019; Ho et al., 2019; Wang et al., 2020) markedly reduces computational complexity while maintaining the ability to capture global context by applying the self-

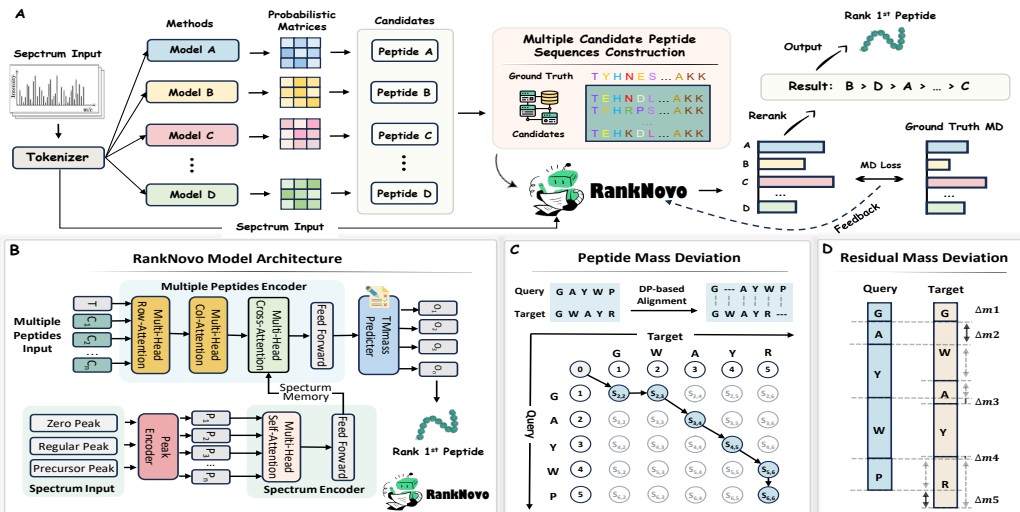

Figure 2: An overview of the RankNovo architecture. (A) Multiple base models generate peptide sequence candidates from the spectrum input, which are subsequently reranked by RankNovo. (B) The architecture of RankNovo incorporates a multi-peptide encoder, utilizing axial attention along both row and column dimensions, and cross-attention to effectively integrate spectrum features. (C) The coarse-grained PMD metric assesses peptide-level differences through dynamic programming-based sequence alignment. (D) The fine-grained RMD metric provides a more granular assessment by capturing residue-level mass deviations between the query and target peptides.

attention mechanism along specific axes of the input data. In the realm of protein modeling, modern deep learning methodologies frequently employ multiple sequence alignments (MSAs) (Feng & Doolittle, 1987) to harness the rich evolutionary and structural information embedded within proteins. For example, the MSA Transformer (Rao et al., 2021), a large-scale protein language model, utilizes axial attention to efficiently process extensive aligned MSA data. Likewise, the prominent protein structure and interaction prediction models AlphaFold2 (Jumper et al., 2021) and AlphaFold3 (Abramson et al., 2024) leverage axial attention to effectively model the input matrix in latent space, thus enabling a broad spectrum of applications in protein modeling and design. Extending these foundational works, our research introduces the first application of the axial attention mechanism to peptide modeling. By examining the similarities and differences between candidates and spectra, our model effectively reranks these candidates. This innovative application expands the utility of axial attention, illustrating its potential in peptide modeling.

## 3 METHOD

### 3.1 PROBLEM FORMULATION

De novo peptide sequencing seeks to deduce the amino acid sequence from a given mass spectrum. Formally, the input set $\mathcal{G} = \{\delta, m^{\text{prec}}, c^{\text{prec}}\}$ is composed of three elements: the spectrum $\delta$, a collection of mass-to-charge ratios (m/z) and intensity signals; the precursor charge $c^{\text{prec}}$, an integer; and the precursor mass $m^{\text{prec}}$, a floating-point value. A spectrum containing $\tilde{k}$ peaks (signal pairs) can be represented as $\{(\lambda_i, \tilde{\mathbf{I}}_i)\}_{i=1}^{\tilde{k}}$. The objective is to identify a set of potential residues, defined as $\mathcal{R} = \{r_1, r_2, \ldots, r_n\}$ providing the input $\mathcal{G}$. The core concept of RankNovo is to integrate multiple relatively weak yet diverse de novo models, and to train the model to select the optimal solution among their outputs. We referred to these models providing candidate predictions as base models.

### 3.2 SPECTRUM AND PEPTIDE EMBEDDING

#### 3.2.1 SPECTRUM EMBEDDING

We filter the peaks in spectrum $\delta_i$ with a m/z range of $[\mu_{\min}, \mu_{\max}]$. Then, the intensities $\tilde{\mathbf{I}}$ of remaining $k$ peaks are square-root transformed and normalized as $\mathbf{I}_i = \frac{\sqrt{\tilde{\mathbf{I}}_i}}{\sum_{j=1}^{k} \sqrt{\tilde{\mathbf{I}}_j}}$.

Following previous works, we use a fixed sinusoidal embedding function $\mathbf{s}^{m/z}$ to project m/z signal into $d$-dimension. Since all $\mu$ falls between $\mu_{min}$ and $\mu_{max}$, we embed the ratio of $\mu$ and $\mu_{min}$ and use $\frac{\mu_{max}}{\mu_{min}}$ as the scale basis of wave length:

$$\mathbf{s}^{m/z}(\mu, i) = \begin{cases} \sin((2\pi\frac{\mu}{\mu_{\min}})/((\frac{\mu_{\max}}{\mu_{\min}})^{\frac{k}{d}})), & \text{if } i = 2k \\ \cos((2\pi\frac{\mu}{\mu_{\min}})/((\frac{\mu_{\max}}{\mu_{\min}})^{\frac{k}{d}})), & \text{if } i = 2k+1 \end{cases} \tag{1}$$

Intensity signals are projected to $d$ dimension with a linear layer because of its relatively lower accuracy, and are summed with corresponding m/z vectors as the initial spectrum embedding $\mathbf{E^0}$, with shape $[k, d]$. Here, no additional positional embeddings are used, as the peaks are inherently unordered in nature.

### 3.2.2 Peptide Candidate Embedding

De novo sequencing is a mass-centric task, thus the prefix and suffix masses of a residual are also embedded in addition to a learnable amino acid embedding, following ContraNovo. Given model dimension $d$, the dimension of learnable embedding, prefix and suffix, denoted as $d_{\text{res}}, d_{\text{prefix}}, d_{\text{suffix}}$, are set to $\frac{d}{2}, \frac{d}{4}$ and $\frac{d}{4}$ respectively. Moreover, the precursor of spectrum are embedded into $d_{\text{prec}}$ ($\frac{d}{2}$) dimension as the sum of $d_{\text{prec}}$-dimensional precursor mass vector and precursor charge vector (Jin et al., 2024). All masses $m$ are embedded using fixed sinusoidal positional embedding:

$$\mathbf{H}^{\mathcal{T}}(m, i) = \begin{cases} \sin\left(\frac{2\pi m}{10000^{k/d_{\mathcal{T}}}}\right), & \text{for } i = 2k \\ \cos\left(\frac{2\pi m}{10000^{k/d_{\mathcal{T}}}}\right), & \text{for } i = 2k+1 \end{cases}, \quad \mathcal{T} \in \{\text{prefix, suffix, prec}\} \tag{2}$$

While the learnable embedding functions for amino acids and precursor charges can be represented as $\mathbf{B}^{\text{res}}$ and $\mathbf{B}^{\text{charge}}$. Then the initial peptide embedding $\mathbf{h}^0 = [\mathbf{h}_{\text{cls}}, \mathbf{h}_1^0, \mathbf{h}_2^0, \ldots, \mathbf{h}_\ell^0]$ can be denoted as:

$$\begin{aligned} \mathbf{h}_i^0 &= \mathbf{B}^{\text{res}}(\text{res}_i) \oplus \mathbf{H}^{\text{prefix}}(m_i^{\text{prefix}}) \oplus \mathbf{H}^{\text{suffix}}(m_i^{\text{suffix}}) \\ \mathbf{h}_{\text{cls}} &= \mathbf{B}^{\text{res}}(\text{cls}) \oplus [\mathbf{H}^{\text{prec}}(m^{\text{prec}}) + \mathbf{B}^{\text{charge}}(c^{\text{prec}})] \oplus \mathbf{0}_{d/2} \end{aligned} \tag{3}$$

where $\oplus$ denotes the concatenation operation over the sequence length dimension and $\mathbf{0}_{d/2}$ denotes learnable 0-quality embedding. RankNovo processes all peptide candidates in a single forward pass. Thus each candidate's embedding is padding to the longest and stacked into the initial MSA embedding $\mathbf{S}^0$. Additionally, a learnable positional embedding is added to each row of $\mathbf{S}^0$, ensuring the model is aware of the token order in a peptide. It is important to note that shuffling the order of rows does not affect the loss prediction due to the absence of column-wise positional embedding.

### 3.3 Accurate Assessment of Peptide Difference

In the context of peptide sequencing tasks, accurate labeling of predictions is crucial for meta-models to effectively identify optimal predictions. Conventional labeling methods for base model predictions, such as binary classification of correctness or edit distance metrics, often lack sufficient precision or fail to account for the mass-centric nature of peptide sequencing. These approaches inadequately capture the nuanced differences between predicted and actual peptide sequences, particularly with respect to amino acid masses, which are fundamental to the sequencing process.

To address these limitations and enhance the reranking of de novo sequencing results, we introduce PMD and RMD, two novel metrics designed for precise quantification of mass differences between peptides. These metrics provides more informative and accurate labels for RankNovo to learn from, thereby improving its discriminative capabilities. PMD and RMD are complementary training objectives during training.

### 3.3.1 Peptide-level Assessment (PMD)

PMD employs a dynamic programming approach analogous to the Needleman-Wunsch algorithm (Needleman & Wunsch, 1970) for sequence alignment, with a specific focus on amino acid masses. Given a set of all possible residues, including both amino acids and post-translational modifications (PTMs), defined as $\mathcal{R} = \{\text{r}_1, \text{r}_2, \ldots, \text{r}_n\}$, where $n$ denotes the number of distinct residue

types, we introduce a corresponding mass look-up table $\mathcal{M} : \mathcal{R} \to \mathbb{R}^+$. Here, $\mathcal{M}(r_i)$ represents the mass of residue $r_i$ for $i \in \{1, 2, ..., n\}$. We define the divergence score matrix $\mathbf{P} \in \mathbb{R}^{n \times n}$, where

$$\mathbf{P}_{i,j} = \begin{cases} 0, & \text{if } i = j \\ |\mathcal{M}(r_i) - \mathcal{M}(r_j)|, & \text{if } i \neq j \end{cases}, \quad i, j \in \{1, 2, \ldots, n\} \tag{4}$$

The gap penalty $\mathbf{g}$ is formalized as the expected symmetric divergence between two distinct residues, given by:

$$\mathbf{g} = \mathbb{E}_{i \neq j} \left[ \mathbf{P}(\mathbf{r}_i, \mathbf{r}_j) \right] = \frac{1}{n(n-1)} \sum_{i=1}^{n} \sum_{\substack{j=1 \\ j \neq i}}^{n} |\mathcal{M}(\mathbf{r}_i) - \mathcal{M}(\mathbf{r}_j)| \tag{5}$$

where $\mathbb{E}_{i \neq j} \left[ \mathbf{P}(\mathbf{r}_i, \mathbf{r}_j) \right]$ is the expectation of the symmetric divergence between the residues.

Given a query peptide sequence $\mathbb{Q} = [\mathbf{r}_{q_1}, \mathbf{r}_{q_2}, \ldots, \mathbf{r}_{q_n}]$ and a target peptide sequence $\mathbb{K} = [\mathbf{r}_{k_1}, \mathbf{r}_{k_2}, \ldots, \mathbf{r}_{k_m}]$, where $n$ and $m$ represent the lengths of the predicted and correct peptides respectively, we initialize a matrix $\mathbf{F} \in \mathbb{R}^{(n+1) \times (m+1)}$. The matrix $\mathbf{F}$ is populated using the following recurrence relation:

$$\mathbf{F}_{i,j} = \begin{cases} 0, & \text{if } i = 1, j = 1 \\ \mathbf{g}(i-1), & \text{if } i \neq 1, j = 1 \\ \mathbf{g}(j-1), & \text{if } i = 1, j \neq 1 \\ \min \left\{ \mathbf{F}_{i-1,j-1} + \mathbf{P}_{q_{i-1},k_{j-1}}, \mathbf{F}_{i-1,j} + \mathbf{g}, \mathbf{F}_{i,j-1} + \mathbf{g} \right\}, & \text{otherwise} \end{cases} \tag{6}$$
$$i \in \{1, 2, \ldots, n+1\}, \quad j \in \{1, 2, \ldots, m+1\}$$

The final output of PMD between the two peptides is computed as $\mathbf{F}_{n+1,m+1}/\mathbf{g}$. Dividing by $\mathbf{g}$ normalizes the value to an order of magnitude around $10^0$, facilitating model fitting. PMD achieves a score of zero only when the predicted peptide exactly matches the correct peptide, making it a precise metric for peptide distance assessment in mass spectrometry-based proteomics.

### 3.3.2 Residual-level Assessment (RMD)

In addition to the peptide-level metric PMD, which the meta-model uses to select the top prediction, we introduce a more fine-grained peptide difference score, RMD. This metric takes advantage of the intrinsic properties of mass spectrometry data. In mass spectrometry, peptide bonds between amino acids are cleaved, generating b- and y-ions. The b-ions, originating from the N-terminus, offer a detailed structural fingerprint of the peptide.

RMD is derived from the prefix masses of the query peptide $\mathbb{Q}$ and the target peptide $\mathbb{K}$, denoted as $\widetilde{\mathbb{Q}} = [\overline{\mathbf{m}}_{q_1}, \overline{\mathbf{m}}_{q_2}, \ldots, \overline{\mathbf{m}}_{q_n}]$ and $\widetilde{\mathbb{K}} = [\overline{\mathbf{m}}_{k_1}, \overline{\mathbf{m}}_{k_2}, \ldots, \overline{\mathbf{m}}_{k_m}]$, where $\overline{\mathbf{m}}_{q_i} = \sum_{j=1}^{i} \mathcal{M}(\mathbf{r}_{q_j})$ and $\overline{\mathbf{m}}_{k_i} = \sum_{j=1}^{i} \mathcal{M}(\mathbf{r}_{k_j})$. This representation is closely aligned with the b-ion mass spectrum. The RMD between these two sequences is represented as a vector $\mathbf{V}$ with $n$ elements, where each element is defined as:

$$\mathbf{V}_i = \overline{\mathbf{m}}_{q_i} - \overline{\mathbf{m}}_{k_{\pi(i)}}, \quad \text{where } \pi(i) = \arg\min_{\tilde{j}} \left| \overline{\mathbf{m}}_{q_i} - \overline{\mathbf{m}}_{k_{\tilde{j}}} \right|. \tag{7}$$

Here, $\pi(i)$ is a learned alignment function that seeks to minimize the mass difference between the $\mathbb{Q}$ and $\mathbb{K}$. By training the model to predict RMD, we encourage it to capture and distinguish subtle structural deviations between peptides. This residual-level task improves the model's ability to identify fine-grained peptide differences, complementing the higher-level insights given by PMD.

### 3.4 Backbone of RankNovo

The backbone of RankNovo need to fulfill three tasks: (1) Extracting spectrum feature,(2) Extracting peptide feature within and among candidates, (3) Mixing spectrum feature and peptide feature to score and rerank peptide candidates.

Spectrum feature extraction can be easily accomplished by a Transformer encoder. After embedding, the initial spectrum representation $\mathbf{E}^0$ is updated by $N_{\text{layer}}$ repetitve self-attention layer:

$$\mathbf{E}^{(i)} = \mathcal{A}_{\text{self}}(\mathbf{E}^{(i-1)}), i = 1, 2, \ldots, N_{\text{layer}} \tag{8}$$

On the other hand, a hybrid peptide track is designed to address tasks (2) and (3) jointly. The peptide track processes the embedded multiple sequence alignment (MSA) feature $\mathbf{S}^0 \in \mathbb{R}^{c \times \ell \times d}$, where $c$ represents the number of candidates, $\ell$ is the sequence length, and $d$ is the model dimension. The final spectrum feature $\mathbf{E}^{N_{\text{layer}}} \in \mathbb{R}^{k \times d}$ is broadcasted across candidates by repeating it to shape $[c, k, d]$, and then integrated into the peptide track. The feature update mechanism is defined as:

$$\mathbf{S}^{(i)} = \mathcal{A}_{\text{cross}} \left( \mathcal{A}_{\text{col}} \left( \mathcal{A}_{\text{row}} (\mathbf{S}^{(i-1)}) \right), \mathbf{E}^{N_{\text{layer}}} \right) \tag{9}$$

where $\mathcal{A}_{\text{row}}$, $\mathcal{A}_{\text{col}}$, and $\mathcal{A}_{\text{cross}}$ denote the row-wise, column-wise, and cross-attention mechanisms, respectively. Here, axial attention is employed to extract peptide features and facilitate information flow between candidate peptides. The iterative application of row and column attention ensures a receptive field that spans the entire $\ell \times k$ token grid, while maintaining a reduced complexity of $\mathcal{O}(c\ell^2 + k^2\ell)$, in contrast to the $\mathcal{O}(c^2\ell^2)$ complexity of standard multi-head self-attention mechanisms. Cross-attention is integrated to incorporate spectrum features into the peptide track, allowing for enhanced alignment between peptides and spectra, and improving overall task performance.

## 3.5 TRAINING WITH JOINT LOSS

The final MSA feature $\mathbf{S}^{N_{\text{layer}}}$ is utilized to predict the PMD and RMD between each candidate peptide and the label peptide. For the peptide-level metric, PMD, the cls token of each candidate is extracted and passed through a linear layer to predict PMD, formulated as: $\text{PMD} = \text{Linear}(\mathbf{h}_{\text{cls}}) \in \mathbb{R}$. Similarly, the $d$-dimensional representation of each amino acid is projected through a linear transformation to predict the residue-level RMD, expressed as: $\text{RMD} = \{\text{Linear}(\mathbf{h}_i^{N_{\text{layer}}}) \in \mathbb{R}\}_{i=1,\dots,\ell}$. Both $\mathcal{L}_{\text{PMD}}$ and $\mathcal{L}_{\text{RMD}}$ are computed using RMSE loss. The optimization objective for training RankNovo is defined as:

$$\mathcal{L} = \lambda \mathcal{L}_{\text{PMD}} + (1 - \lambda) \mathcal{L}_{\text{RMD}} \tag{10}$$

In this work, $\lambda$ is set 0.5 consistently.

## 4 EXPRIMENTS

### 4.1 EXPRIMENT SETUP

**Datasets.** To facilitate a rigorous comparative analysis, we utilized three publicly available peptide-spectrum matches (PSMs) datasets, following the precedent set by recent studies (Yilmaz et al., 2023; Zhang et al., 2024). The MassIVE-KB dataset (Wang et al., 2018) was employed for training, while the 9-species-V1 (Tran et al., 2017) and 9-species-V2 (Yilmaz et al., 2023) datasets were used for evaluation, enabling us to benchmark our model against state-of-the-art de novo peptide sequencing methods. We present a detailed dataset information in Appendix A.1.

**Implementation Details.** RankNovo incorporates six de novo sequencing models, each varying in methodology, as base models during training. These models include Casanovo-V2, ContraNovo, ByNovo, R-Casanovo, R-ContraNovo, and R-ByNovo. Of these, Casanovo-V2 and ContraNovo are directly adopted from the original works and represent both the current and previous state-of-the-art approaches. The latter four models, ByNovo, R-ContraNovo, and R-ByNovo, are developed and trained by ourselves. The detail of base models, traing settings and hyperparameters of RankNovo can be found in Appendix A.2.

**Metrics.** Since reranking task only concerns peptide-level selection, the widely accepted metric peptide recall is our most important metric. Peptide recall is defined as $N_{\text{match}}^{pep}/N_{\text{all}}^{pep}$, here $N_{\text{match}}^{pep}$ is the number of matched peptides and $N_{\text{all}}^{pep}$ is the number of total peptides. The identified peptide is regarded as matched to the label peptide only if every residual is matched. Here residual matching means (1) differing by $< 0.1$ Da in mass and (2) both of the prefix and suffix differing within 0.5 Da. Also, since previous works evaluate model capabilities in residual-level as well, amino acid precision is also taken into concern. Here amino acid precision is defined as $N_{\text{match}}^a/N_{\text{all}}^a$, meaning the percentage of matched residuals among all residuals.

### 4.2 MAIN RESULTS

**Performance on 9-Species-v1 Benchmark Dataset.** The 9-species-V1 dataset, introduced by DeepNovo (Tran et al., 2017), has emerged as a pivotal benchmark for assessing deep learning-based

| | Methods | Bacillus | C. bacteria | Honeybee | Human | M.mazei | Mouse | Rice bean | Tomato | Yeast | **Average** |
|---|---|---|---|---|---|---|---|---|---|---|---|
| | *Amino Acid Precision* | | | | | | | | | | |
| **Baselines** | PEAKS | 0.719 | 0.586 | 0.633 | 0.639 | 0.673 | 0.600 | 0.644 | 0.728 | 0.748 | 0.663 |
| | DeepNovo | 0.742 | 0.602 | 0.630 | 0.610 | 0.694 | 0.623 | 0.679 | 0.731 | 0.750 | 0.673 |
| | PointNovo | 0.768 | 0.589 | 0.644 | 0.606 | 0.712 | 0.626 | 0.730 | 0.733 | 0.779 | 0.687 |
| | Casanovo | 0.749 | 0.603 | 0.629 | 0.586 | 0.679 | 0.689 | 0.668 | 0.721 | 0.684 | 0.667 |
| **Base Models** | Casanovo V2$^\dagger$ | 0.806 | 0.685 | 0.727 | 0.69 | 0.774 | 0.768 | 0.769 | 0.799 | 0.762 | 0.753 |
| | ContraNovo$^\dagger$ | 0.828 | 0.706 | 0.761 | 0.771 | 0.798 | 0.799 | 0.804 | 0.808 | 0.782 | 0.784 |
| | ByNovo$^\star$ | 0.858 | 0.723 | 0.791 | 0.767 | 0.823 | 0.803 | 0.836 | 0.828 | 0.804 | 0.804 |
| | R-Casanovo$^\star$ | 0.804 | 0.699 | 0.728 | 0.719 | 0.769 | 0.776 | 0.782 | 0.795 | 0.762 | 0.759 |
| | R-ContraNovo$^\star$ | 0.839 | 0.716 | 0.775 | 0.782 | 0.806 | 0.811 | 0.816 | 0.822 | 0.798 | 0.796 |
| | R-ByNovo$^\star$ | 0.855 | 0.724 | 0.794 | 0.762 | 0.821 | 0.81 | 0.835 | 0.831 | 0.762 | 0.799 |
| **Ours** | **RankNovo** | **0.874** | **0.746** | **0.81** | **0.802** | **0.84** | **0.828** | **0.859** | **0.844** | **0.816** | **0.824** |
| | *Peptide Recall* | | | | | | | | | | |
| **Baselines** | PEAKS | 0.387 | 0.203 | 0.287 | 0.277 | 0.356 | 0.197 | 0.362 | 0.403 | 0.428 | 0.322 |
| | DeepNovo | 0.449 | 0.253 | 0.330 | 0.293 | 0.422 | 0.286 | 0.436 | 0.454 | 0.462 | 0.376 |
| | PointNovo | 0.518 | 0.298 | 0.396 | 0.351 | 0.478 | 0.355 | 0.511 | 0.513 | 0.534 | 0.439 |
| | Casanovo | 0.537 | 0.330 | 0.406 | 0.341 | 0.478 | 0.426 | 0.506 | 0.521 | 0.490 | 0.448 |
| **Base Models** | Casanovo V2$^\dagger$ | 0.646 | 0.46 | 0.527 | 0.492 | 0.592 | 0.493 | 0.628 | 0.637 | 0.629 | 0.567 |
| | ContraNovo$^\dagger$ | 0.684 | 0.487 | 0.576 | 0.624 | 0.628 | 0.563 | 0.676 | 0.655 | 0.669 | 0.618 |
| | ByNovo$^\star$ | 0.708 | 0.499 | 0.597 | 0.584 | 0.639 | 0.545 | 0.696 | 0.667 | 0.676 | 0.623 |
| | R-Casanovo$^\star$ | 0.628 | 0.467 | 0.515 | 0.511 | 0.57 | 0.505 | 0.611 | 0.611 | 0.601 | 0.558 |
| | R-ContraNovo$^\star$ | 0.682 | 0.499 | 0.583 | 0.606 | 0.621 | 0.566 | 0.673 | 0.654 | 0.664 | 0.616 |
| | R-ByNovo$^\star$ | 0.703 | 0.493 | 0.59 | 0.554 | 0.637 | 0.543 | 0.685 | 0.659 | 0.629 | 0.610 |
| **Ours** | **RankNovo** | **0.738** | **0.539** | **0.63** | **0.642** | **0.672** | **0.583** | **0.733** | **0.691** | **0.703** | **0.660** |

Table 1: Evaluation of RankNovo in comparison to baseline and base methods on the 9-species-v1 test set. Bolded entries indicate the best-performing models. The symbol "†" indicates that the model serves as both a baseline and a base model. "⋆" signifies that the base models were developed and trained by us.

de novo peptide sequencing methods (Yilmaz et al., 2022). In our evaluation, RankNovo exhibits superior performance across all species in the dataset, both at the peptide and amino acid levels. Specifically, RankNovo achieves an average peptide recall of 0.660, surpassing its strongest base model, ByNovo, by 6.1%, and outperforming the current state-of-the-art, ContraNovo, by 4.3%. At the amino acid level, RankNovo reaches a precision of 0.829, outperforming ByNovo by 2.6% and ContraNovo by 4.1%. These results underscore RankNovo's ability to accurately sequence peptides and amino acids across diverse species.

Two key conclusions can be drawn from these results: first, RankNovo establishes a new state-of-the-art in de novo peptide sequencing, surpassing the previous benchmark set by ContraNovo; and second, RankNovo consistently outperforms all of its constituent base models, demonstrating its ability to effectively integrate diverse model outputs, leverage their respective strengths, and mitigate individual weaknesses, thereby reducing generalization error.

**Performance on 9-Species-v2 Benchmark Dataset.** The experimental results in Table 2 clearly indicate that RankNovo consistently outperforms both baseline and comparative models on the 9-Species-v2 dataset, demonstrating superior performance in amino acid precision and peptide recall. Specifically, RankNovo achieves the highest average amino acid precision of 0.906 across all species, with substantial improvements in species such as Bacillus, C. bacteria, and Honeybee. Furthermore, RankNovo attains an average peptide recall of 0.781, outperforming other models across the majority of species, with particularly strong performance in Yeast, Rice bean, and Tomato. These results emphasize the adaptability and effectiveness of RankNovo across a diverse set of species.

| | Methods | Bacillus | C. bacteria | Honeybee | Human | M.mazei | Mouse | Rice bean | Tomato | Yeast | **Average** |
|---|---|---|---|---|---|---|---|---|---|---|---|
| **Amino Acid Precision** | Casanovo V2$^\dagger$ | 0.888 | 0.791 | 0.823 | 0.872 | 0.877 | 0.813 | 0.891 | 0.891 | 0.915 | 0.862 |
| | ContraNovo$^\dagger$ | 0.901 | 0.807 | 0.848 | 0.920 | 0.896 | 0.839 | 0.913 | 0.898 | 0.919 | 0.882 |
| | ByNovo$^\star$ | 0.92 | 0.823 | 0.876 | 0.917 | 0.914 | 0.841 | 0.932 | 0.912 | 0.934 | 0.897 |
| | R-Casanovo$^\star$ | 0.876 | 0.804 | 0.814 | 0.891 | 0.867 | 0.821 | 0.881 | 0.891 | 0.898 | 0.860 |
| | R-ContraNovo$^\star$ | 0.909 | 0.815 | 0.865 | 0.923 | 0.901 | 0.849 | 0.919 | 0.907 | 0.925 | 0.890 |
| | R-ByNovo$^\star$ | 0.919 | 0.822 | 0.879 | 0.912 | 0.912 | 0.843 | 0.932 | 0.913 | 0.936 | 0.897 |
| | **RankNovo** | **0.926** | **0.838** | **0.885** | **0.929** | **0.920** | **0.860** | **0.938** | **0.918** | **0.938** | **0.906** |
| **Peptide Recall** | Casanovo V2$^\dagger$ | 0.793 | 0.558 | 0.669 | 0.712 | 0.754 | 0.555 | 0.772 | 0.783 | 0.837 | 0.714 |
| | ContraNovo$^\dagger$ | 0.815 | 0.575 | 0.711 | 0.820 | 0.780 | 0.616 | 0.799 | 0.794 | 0.854 | 0.752 |
| | ByNovo$^\star$ | 0.833 | 0.582 | 0.731 | 0.789 | 0.799 | 0.596 | 0.814 | 0.807 | 0.871 | 0.758 |
| | R-Casanovo$^\star$ | 0.759 | 0.558 | 0.643 | 0.732 | 0.723 | 0.558 | 0.721 | 0.768 | 0.799 | 0.696 |
| | R-ContraNovo$^\star$ | 0.821 | 0.581 | 0.719 | 0.815 | 0.779 | 0.620 | 0.804 | 0.798 | 0.861 | 0.755 |
| | R-ByNovo$^\star$ | 0.831 | 0.585 | 0.729 | 0.781 | 0.794 | 0.589 | 0.815 | 0.803 | 0.873 | 0.756 |
| | **RankNovo** | **0.851** | **0.620** | **0.752** | **0.820** | **0.813** | **0.629** | **0.836** | **0.822** | **0.885** | **0.781** |

Table 2: Evaluation of RankNovo in comparison to baseline and base methods on the 9-species-v2 test set. Bolded entries indicate the best-performing models. The symbol "†" indicates that the model serves as both a baseline and a base model. "⋆" signifies that the base models were developed and trained by us.

## 4.3 DETAILED ANALYSES

We report average performance on the 9-species-v1 benchmark, with detailed per-species performance in Appendix A.5.

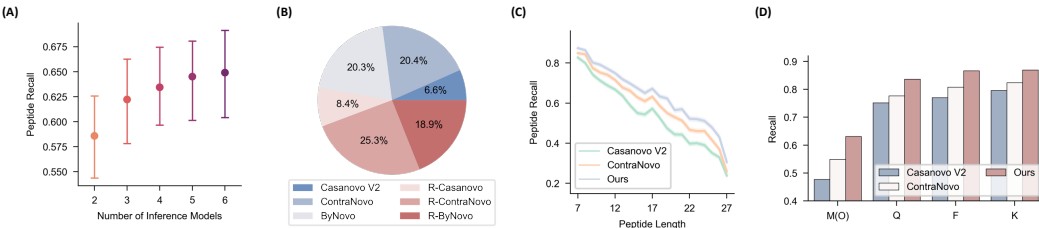

Figure 3: (A) Zero-shot performance of RankNovo when trained on two base models. (B) Unique-correctly selected percentage of base models. (C) Influence of peptide length. (D) The performance comparison of amino acids with similar masses.

**Analysis of Zero-shot Performance.** We demonstrate the zero-shot capability of RankNovo by training it exclusively on predictions from the two lowest-performing base models and progressively incorporating predictions from unseen models into the candidate sets during inference (Figure 3 (A)). The detailed experimental setup is provided in Appendix A.5.1. As the number of inference models increases, the average peptide recall improves, rising from 0.586 with 2 models to 0.649 with 6 models. These results highlight the robust zero-shot capability of RankNovo, demonstrating its efficiency in reranking predictions from models not used during training. This underscores the potential and value of RankNovo for future applications in de novo peptide sequencing.

**Contribution of Each Base Model.** Given the varying capabilities of base models, it is crucial to ensure that each contributes meaningfully to RankNovo's performance. Otherwise, their predictions may introduce unnecessary noise during the reranking process. We analyzed peptide candidates and RankNovo's selections using the Bacillus species data from the 9-species-V1 benchmark. We filtered spectrum samples to retain those where (1) RankNovo's chosen peptide matched the labeled peptide and (2) RankNovo's choice was provided by only one base model. For these filtered samples, we calculated the percentage of times each base model was chosen, using this as a measure of contribution. As illustrated in Figure 3 (B), Casanovo-V2 had the lowest contribution at 6%, while R-ByNovo had the highest at 30%. These results demonstrate that every model contributes to the final performance, as removing any of them would lead to failures on specific test samples.

**Analysis of Peptide Length.** We assess the performance of RankNovo and baselines in recognizing peptides of varying lengths, with a particular emphasis on their effectiveness for both shorter and longer peptides. As shown in Figure 3 (C)), our findings reveal that RankNovo exhibits significantly higher recall compared to ContraNovo for shorter peptides, suggesting enhanced proficiency in recognizing these sequences. As we analyze longer peptides, a discernible trend emerges: the recall for both models shows a downward trajectory, indicative of a decline in recognition capability as peptide length increases. This reduction in performance can be attributed to the heightened complexity associated with longer peptide structures, which may impede model accuracy. Nevertheless, RankNovo consistently outperforms ContraNovo, although the margin of superiority narrows with increasing peptide length.

**Analysis of Amino Acid with Similar Masses.** In de novo peptide sequencing, a sequence is deemed accurately reconstructed only when each residue in the predicted peptide aligns with its corresponding residue in the reference sequence. Prediction accuracy varies across different amino acids, particularly for those with similar masses, which are challenging to distinguish due to nearly overlapping spectral profiles. For instance, oxidized methionine (M(O)) and phenylalanine (F) differ by 0.33 Da, while lysine (K) and glutamine (Q) differ by 0.46 Da. Figure 3 (D))compares RankNovo with two baseline models, Casanovo-V2 and ContraNovo. Utilizing the 9-Species-V1 dataset, recall was computed for each amino acid. Notably, RankNovo achieves an 8.0% improvement in recall for M(O) relative to the baseline models. These results underscore RankNovo's enhanced ability to differentiate between amino acids with closely related masses, effectively capturing subtle mass variations within peptide sequences.

### 4.4 Reranking Framework Comparison and Ablation Study

We compare the performance of RankNovo with other reranking frameworks and conduct ablation studies on its key components. The evaluation is performed on the 9-species-V1 benchmark. Detailed results and analysis are presented in Appendix A.3 and A.4.

| Objective | Avg. Pep. Recall |
|---|---|
| Point-wise | 0.647 |
| Pair-wise | 0.648 |
| List-wise | 0.646 |
| PMD+RMD | **0.660** |

Table 3: Average Peptide Recall on 9-species-V1 test set under the training objective of different reranking loss.

| ID | PMD | RMD | Col-Attn. | Avg. Pep. Recall |
|---|---|---|---|---|
| 1 | | ✔ | ✔ | 0.650 |
| 2 | ✔ | | ✔ | 0.652 |
| 3 | ✔ | ✔ | | 0.653 |
| 4 | ✔ | ✔ | ✔ | **0.660** |

Table 4: Ablation of training metrics combination and column-wise attention modules.

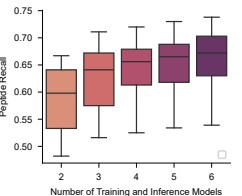

Figure 4: Ablation study of base model combinations.

**Reranking Framework Comparison.** We compare RankNovo with three types of reranking frameworks outlined in RankT5 (Zhuang et al., 2023): point-wise, pair-wise, and list-wise reranking. Following RankT5's methodology, these frameworks are implemented using identical backbone models, differing only in their training objectives. Detailed descriptions are provided in Appendix A.3. As shown in Table 4.4, the three frameworks exhibit comparable performance when reranking de novo sequencing results, achieving approximately 0.647 peptide recall. However, this falls significantly short of the 0.660 recall achieved by RankNovo using our novel metrics, PMD and RMD. These results underscore the specialized efficacy of the RankNovo in peptide sequencing tasks.

**Base Model Combinations Ablation.** We conducted ablation studies to assess the impact of using fewer base models on overall performance. We created five subsets of the final base model set, each containing a different number of base models, and compared the performance of RankNovo when trained and tested with the outputs of these model sets. The selection criteria for these subsets are detailed in Appendix A.4.1. The results, plotted in Figure 4.4, demonstrate a consistent increase in peptide recall as the number of base models increases. This observation supports the hypothesis that a greater diversity of choices leads to improved performance.

**Training Objective Ablation.** Two novel metrics, PMD and RMD, provide the learning objective for RankNovo. The results of experiments 1, 2, and 4 in Table 4.4 demonstrate that the absence of either metric leads to a decrease in peptide recall. The combination of both metrics is necessary to achieve optimal performance.

**Backbone Model Ablation.** The results of experiments 3 and 4 in Table 4.4 reveal a decline in performance without the column-wise attention module, as evidenced by the 0.653 peptide recall after its removal, compared to the original 0.660. This finding supports the hypothesis that incorporating axial attention facilitates the integration of peptide features and contributes to optimal performance.

## 5 Conclusion

In this paper, we introduced RankNovo, a novel list-wise deep reranking framework designed to enhance the accuracy of de novo peptide sequencing under the guidance of our mass deviation metrics, PMD and RMD. RankNovo achieves new state-of-the-art performance on established benchmarks and exhibits strong zero-shot generalization capabilities. The primary limitation of RankNovo lies in the relatively lower inference speed due to the proportional time cost in collecting peptide candidates to the number of base models, potentially constraining its application in scenarios where rapid sequencing is crucial. Future work could explore efficient candidate sampling methods, such as utilizing base models with partially shared weights to reduce computational overhead. Additionally, investigating the scalability of the reranking framework by expanding the candidate pool size presents an interesting research direction.

Despite the speed constraints, RankNovo represents the first deep reranking framework to offer a flexible trade-off between inference time and performance, introducing a novel perspective for performance enhancement. We anticipate that, influenced by RankNovo, future algorithms in this field will benefit from the synergistic approach of simultaneously improving single-model performance and developing advanced reranking strategies.

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

# A  APPENDIX

## A.1  DATASET DETAILS

MassIVE-KB is a widely used training dataset employed in previous studies such as GLEAMS (Bittremieux et al., 2022) and CasaNovo. Its popularity stems from its substantial size, comprising 30 million PSMs, and its diverse distribution, characterized by sources from different instruments and a rich variety of posttranslational modifications. The 9-species-V1 dataset, introduced by DeepNovo, contains approximately 1.5 million PSMs with a database search false discovery rate of 1%. These PSMs are derived from nine distinct experiments conducted on the same instrument but analyzing peptides from various species, ensuring the dataset's diversity. The 9-species-V2 dataset, an updated version of 9-species-V1 collected in CasaNovo-V2, contains 2.8 million PSMs. Building upon V1, V2 was refined using the Crux protein identification tool (McIlwain et al., 2014) and filtered with a Percolator (Spivak et al., 2009) q-value < 0.01, enhancing its quality.

In addition to the PSMs datasets, we collected the best peptide predictions from six baseline models: Casanovo, ContraNovo, ByNovo, Re-Casanovo, Re-ContraNovo, and Re-ByNovo for each spectrum. Due to the substantial size of the MassIVE-KB training dataset and the computational constraints of beam search, we employed greedy decoding for peptide collection in the training phase. Additionally, spectrums which are correctly predicted by all the six base models are excluded, which remains 7 million spectrums for the training set. Conversely, for the evaluation datasets (9-species-V1 and 9-species-V2), we utilized beam search decoding with a beam size of 5. This approach aligns with previous works and enables optimal benchmark performance during evaluation.

## A.2  IMPLEMENTATION DETAILS

### A.2.1  HYPERPARAMETERS

RankNovo is implemented with the following hyperparameters: 8 layers for both the spectrum encoder and peptide feature mixer, 8 attention heads, a model dimension of 512, a feed-forward dimension of 1024, and a dropout rate of 0.30.

For spectrum and peptide preprocessing, spectra are filtered according to the following criteria: minimum m/z ratio of 50.5 Da, maximum m/z ratio of 4500.0 Da, maximum peak number of 300, precursor m/z tolerance of 2.0 Da, and precursor mass tolerance of 50 ppm. Spectra with more than 300 peaks are truncated, retaining only the 300 peaks with the highest intensities. Spectra that do not satisfy the precursor m/z tolerance and precursor mass tolerance are removed. Additionally, peptides longer than 100 amino acids are truncated. During evaluation, all base models generate peptides using a beam search with a size of 5.

RankNovo is trained using an AdamW optimizer with a learning rate of 1e-4 and weight decay of 8e-5. The model is trained with a batch size of 256 for 5 epochs, including a 1-epoch warm-up period. A cosine learning rate scheduler is employed, and gradients are clipped to 1.5 using L2 norm. The training is conducted on 4 A100 40G GPUs.

### A.2.2  BASELINES

Our benchmark evaluation first compares RankNovo with its base model components to assess the effectiveness of the reranking framework. The components include Casanovo-V2, ContraNovo, ByNovo, R-Casanovo, R-ContraNovo, and R-ByNovo, which collectively represent both current and previous state-of-the-art models for de novo sequencing, particularly ContraNovo and Casanovo-V2. For consistency with prior work, we also evaluate four additional benchmark algorithms: DeepNovo, PointNovo, Casanovo-V1, and PEAKS. Notably, PEAKS employs a dynamic programming-based approach, while the remaining three are deep learning-based models.

### A.2.3  BASE MODEL SELECTION

The selection of base models decides the performance upper bound of RankNovo. The selection of base models should follow three criterions: (1) The training datasets of each base model should have no interset with the test dataset, which is a common data leakage problem in ensemble learning. (2) Base models should be diverse in data preference.

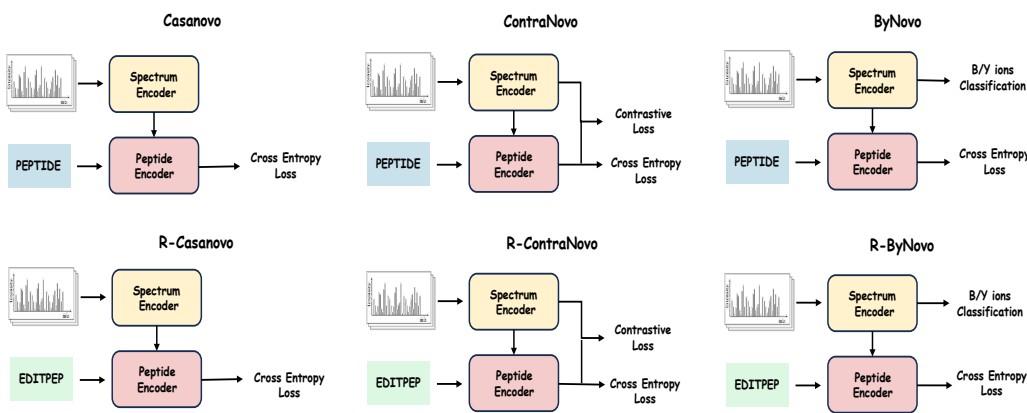

Figure 5: The architecture of six base models for de novo peptide sequencing.

In our research, we selected six models for de novo peptide sequencing as our base models: Casanovo (Yilmaz et al., 2022; 2023), ContraNovo (Jin et al., 2024), ByNovo, R-Casanovo, R-ContraNovo, and R-ByNovo as show in Figure 5. All these models are Transformer-based, but each employs different methodologies. Casanovo and ContraNovo are based on previous work, and we directly use the official checkpoints for these models. The latter four models, ByNovo, R-ContraNovo, and R-ByNovo, are developed and trained by ourselves:

1. **CasaNovo V2**: This fundamental Transformer-based de novo sequencing model treats de novo sequencing as a sequence-to-sequence machine translation task, translating spectra into peptides.

2. **ContraNovo**: This model leverages contrastive learning to enhance feature extraction. It also more effectively utilizes amino acid masses, as well as prefix and suffix masses, during the encoding and decoding processes.

3. **ByNovo**: Developed in-house to increase model diversity, ByNovo incorporates the prediction of BY ions using the output of the spectrum encoder as an auxiliary task. Refer to Appendix A.2.4 for ByNovo implementation details.

4. **R-Casanovo**: Inspired by recent studies (Eloff et al., 2023; Wu et al., 2023), this model trains to decode peptide sequences in reverse. The one-way nature of auto-regressive decoding leads to different results when the sequence is decoded in reverse. R-Casanovo is the reverse-decoding version of Casanovo.

5. **R-ContraNovo**: The reverse decoding version of ContraNovo.

6. **R-ByNovo**: The reverse decoding version of ByNovo.

To avoid data leakage problem, these six models are all trained on Massive-KB, the largest available de novo peptide sequencing dataset in public, and are evaluated in a zero-shot manner on Nine-Species and Nine-Species V2 dataset, following previous works. On the other hand, the difference in methodologies successfully leads to the variety in data preference of base models (Fig 1). These settings meet the criterias for base model selection. The architectures of these six models in detail can be found in the Appendix.

### A.2.4 BYNOVO DETAILS

ByNovo introduces an auxiliary task for identifying key ion PEAKS alongside the primary peptide sequencing task. Real-world spectral data often contain numerous noise PEAKS and ion PEAKS that are weakly related or unrelated to de novo sequencing, which can interfere with the model's performance (Breci et al., 2003; Tabb et al., 2003). To mitigate this issue, ByNovo incorporates ion peak identification as an auxiliary objective, modeling ion recognition as a token-level classification problem by introducing an ion annotation head within the encoder.

Specifically, ByNovo first labels the ion PEAKS in the spectrum, assigning an ion type $l_i \in \{b/y, other\}$ to each peak. The task is formalized as maximizing the conditional probability of predicting the ion type $l_i$, given the mass-to-charge ratio $m_i$ and charge $z_i$, as shown in equation:

$$\mathbf{P}(l_i \mid m_i, z_i, \theta) = \frac{e^{f(m_i, z_i, l_i; \theta)}}{\sum_{l' \in L} e^{f(m_i, z_i, l'; \theta)}} \quad (11)$$

where $f(\cdot)$ is the classification model, and $\theta$ denotes the model parameters. For the entire ion peak sequence in a spectrum, ByNovo maximizes the joint probability, as defined in equation:

$$\mathbf{P}(\mathbf{L} \mid \mathbf{S}) = \prod_{i=1}^{N} \mathbf{P}(l_i \mid m_i, z_i, \theta) \quad (12)$$

In this classification task, ByNovo uses Focal Loss as the supervision loss function, defined in equation:

$$\mathcal{L}_{\text{ion}}(p_t) = -(1 - p_t)^\gamma \log(p_t) \quad (13)$$

where $p_t$ is the predicted probability of the correct class, and $\gamma > 0$ is a focusing parameter. Focal Loss assigns smaller penalties to well-classified examples with high confidence, while increasing the loss for hard-to-classify samples. This encourages the model to focus on learning difficult examples, which is beneficial for detecting easily overlooked b/y ion PEAKS in spectra.

By explicitly supervising ion peak classification as a token classification task, the model is guided to learn the critical features distinguishing b/y ions from other ions. In complex spectral scenarios, this supervision signal implicitly constrains and regularizes the peptide sequence prediction process, improving sequencing accuracy. This multi-task learning approach helps the model learn more discriminative feature representations, reduces the risk of overfitting, and enhances generalization performance.

### A.3 Reranking Framework Comparison

| | Objective | Col-Attn | Bacillus | C. bacteria | Honeybee | Human | M.mazei | Mouse | Rice bean | Tomato | Yeast | Average |
|---|---|---|---|---|---|---|---|---|---|---|---|---|
| **Amino Acid Precision** | Point | ✗ | 0.869 | 0.741 | 0.803 | 0.796 | 0.835 | 0.824 | 0.85 | 0.841 | 0.811 | 0.819 |
| | Pair | ✗ | 0.87 | 0.741 | 0.803 | 0.799 | 0.836 | 0.826 | 0.854 | 0.841 | 0.811 | 0.82 |
| | List | ✗ | 0.865 | 0.737 | 0.799 | 0.786 | 0.829 | 0.825 | 0.832 | 0.844 | 0.817 | 0.815 |
| | PMD+RMD | ✗ | 0.871 | 0.745 | 0.809 | 0.801 | 0.839 | 0.828 | 0.854 | 0.844 | 0.812 | 0.822 |
| | Point | ✔ | 0.871 | 0.74 | 0.806 | 0.8 | 0.837 | 0.825 | 0.852 | 0.841 | 0.812 | 0.82 |
| | Pair | ✔ | 0.871 | 0.742 | 0.804 | 0.798 | 0.837 | 0.825 | 0.853 | 0.842 | 0.812 | 0.82 |
| | List | ✔ | 0.866 | 0.738 | 0.797 | 0.780 | 0.832 | 0.826 | 0.846 | 0.845 | 0.821 | 0.817 |
| | **PMD+RMD**† | ✔ | **0.874** | **0.746** | **0.81** | **0.802** | **0.84** | **0.828** | **0.859** | **0.844** | **0.816** | **0.824** |
| **Peptide Recall** | Point | ✗ | 0.727 | 0.528 | 0.614 | 0.628 | 0.660 | 0.575 | 0.721 | 0.680 | 0.690 | 0.647 |
| | Pair | ✗ | 0.728 | 0.529 | 0.614 | 0.631 | 0.661 | 0.573 | 0.722 | 0.681 | 0.692 | 0.648 |
| | List | ✗ | 0.725 | 0.540 | 0.622 | 0.613 | 0.663 | 0.590 | 0.694 | 0.670 | 0.705 | 0.646 |
| | PMD+RMD | ✗ | 0.732 | 0.535 | 0.623 | 0.638 | 0.664 | 0.579 | 0.727 | 0.685 | 0.695 | 0.653 |
| | Point | ✔ | 0.731 | 0.529 | 0.619 | 0.634 | 0.663 | 0.577 | 0.722 | 0.681 | 0.693 | 0.65 |
| | Pair | ✔ | 0.729 | 0.531 | 0.616 | 0.634 | 0.662 | 0.575 | 0.721 | 0.684 | 0.696 | 0.65 |
| | List | ✔ | 0.728 | 0.536 | 0.613 | 0.605 | 0.663 | 0.588 | 0.710 | 0.694 | 0.706 | 0.649 |
| | **PMD+RMD**† | ✔ | **0.738** | **0.539** | **0.63** | **0.642** | **0.672** | **0.583** | **0.733** | **0.691** | **0.703** | **0.660** |

Table 5: Performance comparison on 9-species-V1 test set when the reranking framework varies. The symbol "†" indicates that the model is the final RankNovo mentioned in the main text.

Our RankNovo reranking framework features using the accurate peptide mass deviation metric PMD and RMD as similarity labels. Additionally, we use axial attention (particularly column-wise attention compared to ordinary language models) to boost peptide feature mixing. Here we compare this framework to some established reranking settings in order to prove that RankNovo captures the key modality feature of peptide and mass spectrums and is a superior methodology than those used in NLP tasks on peptide sequencing task.

Our comparison involves two aspects: the reranking loss level and backbone model level. We mainly compare RankNovo with the classic RankT5 (Zhuang et al., 2023) framework. RankT5 summurized the three common style of reranking losses: point-wise, pair-wise and list-wise loss. Suppose a given query $q_i$ has $N$ potentially relevant candidate documents $d_{i1}, d_{i2}, \ldots, d_{iN}$ to rerank, a reranking framework uses a backbone language model $\mathcal{M}$ to extract the latent $z_{ij}$ representing the relationship

between $q_i$ and $d_{ij}$. Then, $z_{ij}$ is projected (in RankT5 the 'projection' is accomplished by learning a new word) to predict the similarity score $\hat{y}_{ij}$. The process can be summarized as:

$$\hat{y}_{ij} = \text{Projection}(\mathcal{M}(q_i, d_{ij})) \tag{14}$$

For peptide sequencing task, we set the true relevance label $y_{ij}$ as a binary classification label (since our metric PMD and RMD is not adopted). Then the point-wise loss function for each sample $q_i$ equals a sumation of binary cross entropy (BCE) losses between each query-document pair.

$$\mathcal{L}_{\text{Point}}(y_i, \hat{y}_i) = - \sum_{j|y_{ij}=1} log(\sigma(\hat{y}_{ij})) - \sum_{j|y_{ij}=0} log(\sigma(1 - \hat{y}_{ij})), \sigma(x) = \frac{1}{1 + e^{-x}} \tag{15}$$

Pair-wise reranking loss focuses on enlarging the predicted similarity deviation between relevant query-document pairs and the irrelevant ones, which can be represented as:

$$\mathcal{L}_{\text{Pair}}(y_i, \hat{y}_i) = \sum_{j=1}^{N} \sum_{j'=1}^{N} \mathbb{I}_{y_{ij} > y_{ij'}} log(1 + e^{\hat{y}_{ij'} - \hat{y}_{ij}}) \tag{16}$$

List-wise loss views reranking as a $N$-class classfication, thus the loss function can be represented as:

$$\mathcal{L}_{\text{List}}(y_i, \hat{y}_i) = - \sum_{j=1}^{N} y_{ij} log(\frac{e^{\hat{y}_{ij}}}{\sum_{j'} e^{\hat{y}_{ij'}}}) \tag{17}$$

For backbone model $\mathcal{M}$, the encoder-decoder framework is inarguable. The only concern is whether the candidates $d_{ij}$ should be able to 'see' each other. Some pair-wise or list-wise reranking work uses paired candiates input and a post-ranking procedure. Here we uses column-wise attention modules to enable list-level perception field because the existing methods for enabling communications between candidates are too diverse, making exhaustive comparison unreal.

Detailed results can be found in Table 5. Whether using column-wise attention or not, the best model among point-wise, pair-wise and list-wise framework falls behind at least 0.1 than RankNovo in terms of peptide recall, which achieves an average of 0.660 peptide recall across the nine species. Therefore, RankNovo is more suitable for sequencing task than common NLP reranking frameworks. It's worth noticing that amino acid recall and peptide precision not necessarily follow the same trend, especially when the peptide recalls between two models are close, because different models may solve tasks of varying lengths. However in peptide sequencing task, the prime concern is whether a spectrum can be identified. Therefore our analysis focuses on peptide recall, so as in Appendix A.4.

## A.4 ABLATION STUDY

### A.4.1 BASE MODEL CONTRIBUTION ABLATION

The six base models of RankNovo are Casanovo-V2, ContraNovo, ByNovo, R-Casanovo, R-ContraNovo and R-ByNovo. In this section, we would like to examine the necessity of each base model to achieve optimal performance, both during training and inference.

| Model Num. | Base Model Set |
|---|---|
| 2 | Casanovo-V2, R-Casanovo |
| 3 | Casanovo-V2, R-Casanovo, R-ByNovo |
| 4 | Casanovo-V2, R-Casanovo, R-ByNovo, R-ContraNovo |
| 5 | Casanovo-V2, R-Casanovo, R-ByNovo, R-ContraNovo, ContraNovo |
| 6 | Casanovo-V2, R-Casanovo, R-ByNovo, R-ContraNovo, ContraNovo, ByNovo |

Table 6: Description of different combination of base models

The six base models of RankNovo have dozens of subsets, thus it's impossible to study every combination. Here in order to study the influence of the number of base models and each model, we select five subsets. Since the performance of these six models on 9-species-V1 from poor to strong is:

Casanovo-V2, R-Casanovo, R-ByNovo, R-ContraNovo, ContraNovo and ByNovo, the five subsets are formed by sequentially removing the strongest model, until reaching a minimum model number of 2. The details of these five combinations are listed in Table 6

| | N. Train | N. Infer | Bacillus | C. bacteria | Honeybee | Human | M.mazei | Mouse | Rice bean | Tomato | Yeast | Average |
|---|---|---|---|---|---|---|---|---|---|---|---|---|
| **Amino Acid Precision** | 2 | 2 | 0.832 | 0.721 | 0.752 | 0.735 | 0.795 | 0.785 | 0.804 | 0.806 | 0.796 | 0.781 |
| | 3 | 3 | 0.864 | 0.749 | 0.8 | 0.774 | 0.828 | 0.811 | 0.846 | 0.829 | 0.796 | 0.811 |
| | 4 | 4 | 0.865 | 0.751 | 0.802 | 0.794 | 0.834 | 0.819 | 0.847 | 0.831 | 0.817 | 0.818 |
| | 5 | 5 | 0.87 | 0.756 | 0.804 | 0.802 | 0.835 | 0.825 | 0.852 | 0.832 | 0.822 | 0.822 |
| | 6† | 6 | **0.874** | **0.746** | **0.81** | **0.802** | **0.84** | **0.828** | **0.859** | **0.844** | **0.816** | **0.824** |
| **Peptide Recall** | 2 | 2 | 0.667 | 0.482 | 0.548 | 0.533 | 0.598 | 0.519 | 0.649 | 0.641 | 0.636 | 0.586 |
| | 3 | 3 | 0.711 | 0.516 | 0.599 | 0.575 | 0.645 | 0.55 | 0.701 | 0.672 | 0.641 | 0.623 |
| | 4 | 4 | 0.72 | 0.525 | 0.613 | 0.613 | 0.656 | 0.568 | 0.711 | 0.679 | 0.673 | 0.64 |
| | 5 | 5 | 0.73 | 0.534 | 0.618 | 0.633 | 0.665 | 0.583 | 0.724 | 0.683 | 0.688 | 0.651 |
| | 6† | 6 | **0.738** | **0.539** | **0.63** | **0.642** | **0.672** | **0.583** | **0.733** | **0.691** | **0.703** | **0.660** |

Table 7: Peptide recall evaluation of RankNovo on 9-species-V1 test set when the training base model set and the inference base model set are the same and vary. The symbol "†" indicates that the model is the final RankNovo mentioned in the main text.

Firstly we consider the impact of some base models being completely disregarded, during training and inference. From Table 7, we can see that can more base models are used, the peptide recall on 9-species-V1 dataset ascends, from the lowest 0.586 of two models to the highest 0.660 of six models. On the other hand, even when all the six models are used during inference, the absence of models during training affects the final performance. As in Table 8, the combination of two base models achieve the lowest peptide recall of 0.649, 1.6% worse than the combination of all models.

| | N. Train | N. Infer | Bacillus | C. bacteria | Honeybee | Human | M.mazei | Mouse | Rice bean | Tomato | Yeast | Average |
|---|---|---|---|---|---|---|---|---|---|---|---|---|
| **Amino Acid Precision** | 2 | 6 | 0.870 | 0.754 | 0.806 | 0.803 | 0.837 | 0.821 | 0.856 | 0.833 | 0.826 | 0.823 |
| | 3 | 6 | 0.872 | 0.757 | 0.805 | 0.8 | 0.835 | 0.822 | 0.852 | 0.834 | 0.826 | 0.823 |
| | 4 | 6 | 0.872 | 0.756 | 0.807 | 0.805 | 0.839 | 0.822 | 0.855 | 0.835 | 0.827 | 0.824 |
| | 5 | 6 | 0.875 | 0.761 | 0.81 | 0.806 | 0.839 | 0.826 | 0.858 | 0.837 | 0.828 | 0.827 |
| | 6† | 6 | **0.874** | **0.746** | **0.81** | **0.802** | **0.84** | **0.828** | **0.859** | **0.844** | **0.816** | **0.824** |
| **Peptide Recall** | 2 | 6 | 0.727 | 0.525 | 0.613 | 0.627 | 0.663 | 0.577 | 0.723 | 0.684 | 0.686 | 0.647 |
| | 3 | 6 | 0.731 | 0.533 | 0.617 | 0.628 | 0.661 | 0.579 | 0.719 | 0.685 | 0.693 | 0.649 |
| | 4 | 6 | 0.733 | 0.533 | 0.625 | 0.638 | 0.668 | 0.580 | 0.729 | 0.688 | 0.700 | 0.655 |
| | 5 | 6 | 0.734 | 0.537 | 0.624 | 0.638 | 0.67 | 0.584 | 0.729 | 0.689 | 0.696 | 0.657 |
| | 6† | 6 | **0.738** | **0.539** | **0.63** | **0.642** | **0.672** | **0.583** | **0.733** | **0.691** | **0.703** | **0.660** |

Table 8: Peptide recall evaluation of RankNovo on 9-species-V1 test set when the training base model set varies and the inference base model set is fixed. The symbol "†" indicates that the model is the final RankNovo mentioned in the main text.

Combining these two experiments, two important results are shown. Firstly, the impact of not using all models exists, both at training or at inference stage. These means the integral of more diversity during training enrich the knowledge of RankNovo. Secondly, we can see that the number of models during inference is more important than that during training. Both training with two models, the peptide recall raises by 10.7% when the number of inference models increases from 2 to 6. This shows RankNovo's zero-shot generalization ability, which is more delicately shown in Section A.5.1.

### A.4.2 TRAINING OBJECTIVE ABLATION

| | Objective | Bacillus | C. bacteria | Honeybee | Human | M.mazei | Mouse | Rice bean | Tomato | Yeast | Average |
|---|---|---|---|---|---|---|---|---|---|---|---|
| **Amino Acid Precision** | RMD | 0.869 | 0.755 | 0.807 | 0.802 | 0.838 | 0.822 | 0.853 | 0.834 | 0.827 | 0.821 |
| | PMD | 0.871 | 0.742 | 0.810 | 0.806 | 0.836 | 0.823 | 0.856 | 0.835 | 0.821 | 0.822 |
| | **PMD + RMD†** | **0.874** | **0.746** | **0.81** | **0.802** | **0.84** | **0.828** | **0.859** | **0.844** | **0.816** | **0.824** |
| **Peptide Recall** | RMD | 0.731 | 0.529 | 0.618 | 0.632 | 0.664 | 0.576 | 0.723 | 0.684 | 0.691 | 0.65 |
| | PMD | 0.731 | 0.534 | 0.623 | 0.637 | 0.664 | 0.577 | 0.726 | 0.685 | 0.694 | 0.652 |
| | **PMD + RMD†** | **0.738** | **0.539** | **0.63** | **0.642** | **0.672** | **0.583** | **0.733** | **0.691** | **0.703** | **0.660** |

Table 9: Evaluation of performance on 9-species-V1 test set when training under different objective. The symbol "†" indicates that the model is the final RankNovo mentioned in the main text.

In this work, we introduces two novel metrics, PMD and RMD, as the learning objective of reranking models. Here we conduct the ablation study of the effect of the combinational use of these two metrics. As shown in Table 9, uses RMD alone achieves the lowest peptide recall of 0.650, while only using PMD alone is better, with a peptide recall of 0.657. The best ppetide recall of 0.660 is achieved when both PMD and RMD are used.

### A.4.3 MODEL ARCHITECTURE ABLATION

| | Col-Attn | Bacillus | C. bacteria | Honeybee | Human | M.mazei | Mouse | Rice bean | Tomato | Yeast | Average |
|---|---|---|---|---|---|---|---|---|---|---|---|
| **AA Precision** | ✗ | 0.871 | 0.745 | 0.809 | 0.801 | 0.839 | 0.828 | 0.854 | 0.844 | 0.812 | 0.822 |
| | ✔† | **0.874** | **0.746** | **0.81** | **0.802** | **0.84** | **0.828** | **0.859** | **0.844** | **0.816** | **0.824** |
| **Peptide Recall** | ✗ | 0.732 | 0.535 | 0.623 | 0.638 | 0.664 | 0.579 | 0.727 | 0.685 | 0.695 | 0.653 |
| | ✔† | **0.738** | **0.539** | **0.63** | **0.642** | **0.672** | **0.583** | **0.733** | **0.691** | **0.703** | **0.660** |

Table 10: Performance comparison of RankNovo on 9-species-V1 test set between using column-wise attention in the peptide feature mixer or not. The symbol "†" indicates that the model is the final RankNovo mentioned in the main text.

The effect of whether using column-wise attention has already been mentioned in Section A.3. In this section we emphasize its effect when the training objective is chosen to be the combination of PMD and RMD. From Table 10 we can see when using column-wise attention modules, the average peptide recall across the nine species rises from 0.657 to 0.660. This shows column-wise attention's contribution to optimal performance of RankNovo.

### A.5 ADDITIONAL RESULTS

### A.5.1 ANALYSIS OF ZERO-SHOT PERFORMANCE

| | N. Train | N. Infer | Bacillus | C. bacteria | Honeybee | Human | M.mazei | Mouse | Rice bean | Tomato | Yeast | Average |
|---|---|---|---|---|---|---|---|---|---|---|---|---|
| **Amino Acid Precision** | 2 | 2 | 0.832 | 0.721 | 0.752 | 0.735 | 0.795 | 0.785 | 0.804 | 0.806 | 0.796 | 0.781 |
| | 2 | 3 | 0.86 | 0.732 | 0.799 | 0.773 | 0.826 | 0.813 | 0.845 | 0.835 | 0.778 | 0.807 |
| | 2 | 4 | 0.861 | 0.733 | 0.798 | 0.786 | 0.828 | 0.82 | 0.845 | 0.837 | 0.804 | 0.812 |
| | 2 | 5 | 0.864 | 0.737 | 0.801 | 0.797 | 0.832 | 0.825 | 0.849 | 0.839 | 0.807 | 0.817 |
| | **2** | **6** | **0.873** | **0.757** | **0.809** | **0.806** | **0.840** | **0.824** | **0.859** | **0.836** | **0.829** | **0.826** |
| **Peptide Recall** | 2 | 2 | 0.667 | 0.482 | 0.548 | 0.533 | 0.598 | 0.519 | 0.649 | 0.641 | 0.636 | 0.586 |
| | 2 | 3 | 0.707 | 0.51 | 0.596 | 0.579 | 0.642 | 0.551 | 0.705 | 0.671 | 0.64 | 0.622 |
| | 2 | 4 | 0.716 | 0.518 | 0.605 | 0.604 | 0.653 | 0.567 | 0.709 | 0.677 | 0.662 | 0.635 |
| | 2 | 5 | 0.722 | 0.525 | 0.611 | 0.628 | 0.661 | 0.579 | 0.72 | 0.681 | 0.68 | 0.645 |
| | **2** | **6** | **0.729** | **0.527** | **0.615** | **0.629** | **0.665** | **0.579** | **0.725** | **0.686** | **0.688** | **0.649** |

Table 11: Zero-shot performance of a fix training base model set of two models on unseen models. The numbers are calculated on 9-species-V1 dataset.

We demonstrate the zero-shot capability of RankNovo by training it exclusively on predictions from the two lowest-performing base models and progressively incorporating predictions from unseen models into the candidate sets for each spectrum during inference. As shown in Table 11, as the number of inference models increases, the average peptide recall improves, rising from 0.586 with 2 models to 0.649 with 6 models.

### A.5.2 ANALYSIS OF AMINO ACID IDENTIFICATION WITH SIMILAR MASSES

The experimental results across the nine species, as illustrated in Figure 6, exhibit a consistent improvement in recall for key amino acids (M(O), Q, F, K) when leveraging RankNovo over the baseline methods, Casanovo V2 and ContraNovo. RankNovo consistently outperforms the baselines across all species, particularly in M(O) and F. The recall improvements are most pronounced in species like yeast, ricebean, and honeybee, where RankNovo demonstrates significant performance gains. These results emphasize the strong generalization capabilities of RankNovo across diverse species and its effectiveness in addressing the ambiguities introduced by amino acids with similar masses. The consistent superiority of RankNovo underscores its potential to advance peptide sequencing, especially within complex biological datasets.

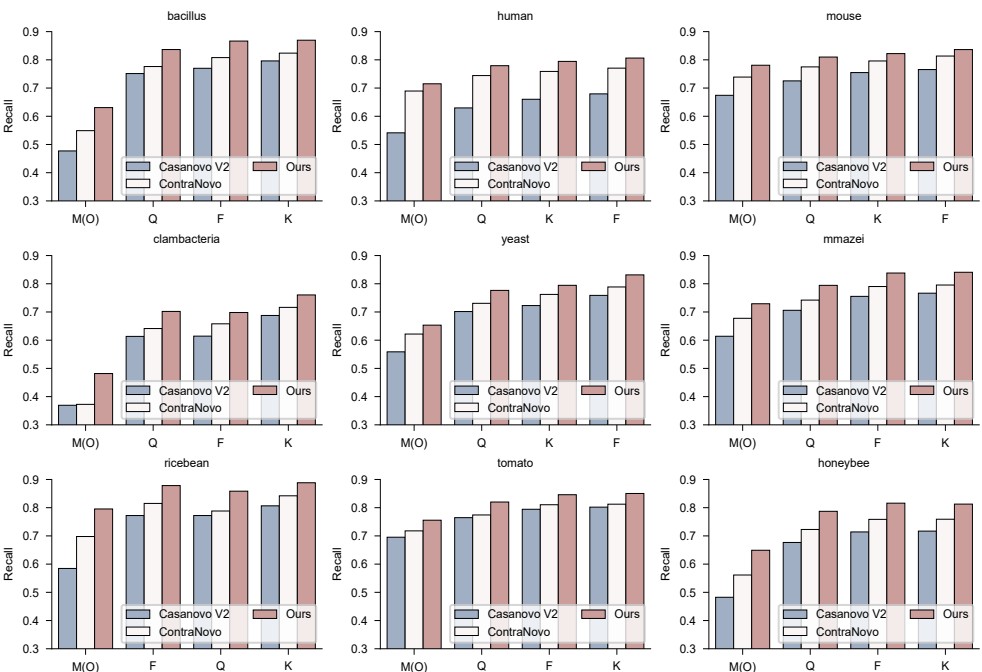

Figure 6: The performance comparison of amino acids with similar masses. The numbers are calculated on 9-species-V1 dataset.

### A.5.3 ANALYSIS OF PEPTIDE LENGTH

The results in Figure 7 demonstrate that our model consistently surpasses the baseline methods, Casanovo V2 and ContraNovo, across a wide variety of species. Specifically, for shorter peptides (lengths 7 to 17), our model achieves significantly higher recall across all species, underscoring its enhanced capacity to capture key sequence patterns in simpler peptide structures. As peptide length increases, performance across all models declines progressively, indicating that longer peptides introduce additional structural complexity that impairs recognition accuracy. Nonetheless, our model maintains a competitive advantage, consistently outperforming the baselines for most species. However, the performance gap diminishes as peptide length increases, likely due to the heightened challenges associated with recognizing longer sequences. These results highlight the effectiveness of our model in processing peptides of varying lengths, as well as its strong generalization capability across diverse species.

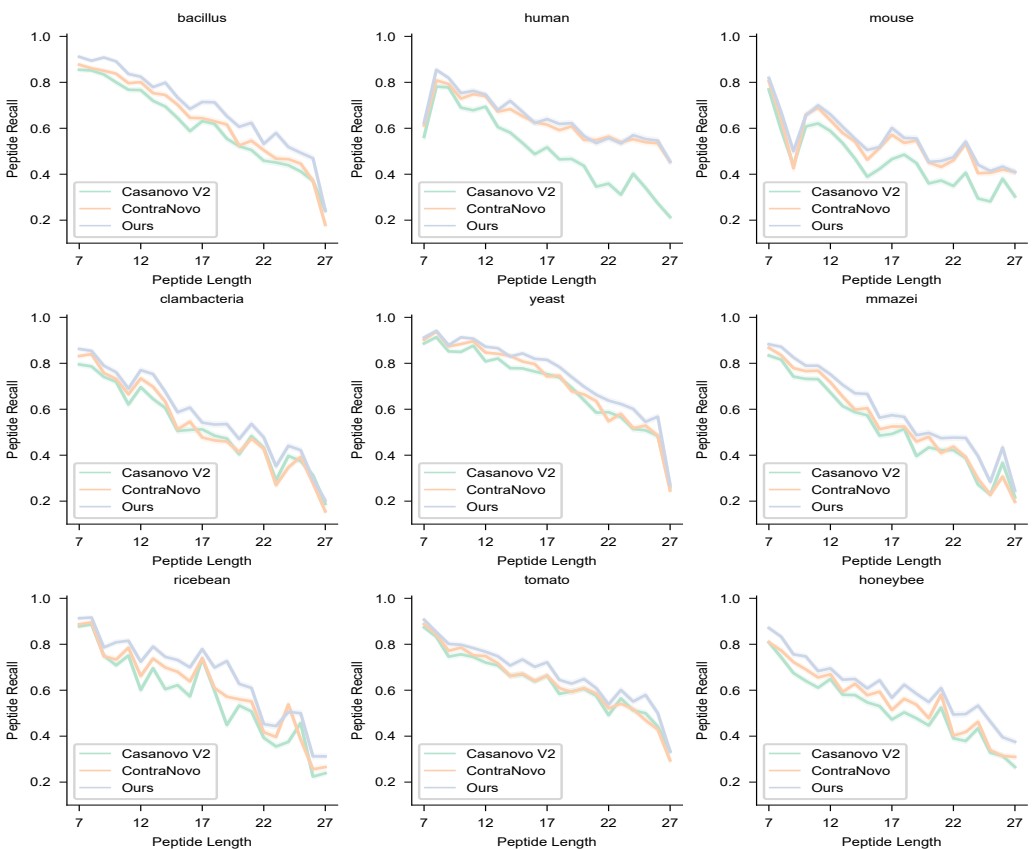

Figure 7: Influence of peptide length on 9-species-V1 dataset.

### A.5.4 CONTRIBUTION OF EACH BASE MODEL

By analyzing the contributions of individual base models across nine species, we uncover distinct patterns of efficacy, as depicted in Figure 8. Each base model exhibits varying degrees of influence on RankNovo's peptide selection, underscoring their complementary strengths. Notably, R-ByNovo consistently demonstrates the highest contribution in most species, reaching 41.7% in yeast, while Casanovo-V2 contributes less significantly, particularly in species like tomato and mouse. This variation suggests that different models capture species-specific features with varying effectiveness. The consistent, albeit variable, contributions of each base model highlight the critical importance of model diversity; removing any single model would likely degrade performance for certain species. These findings illustrate the robustness of the ensemble approach, where integrating multiple models compensates for the limitations of individual ones, enabling RankNovo to generalize effectively across a broad range of species and peptide structures.

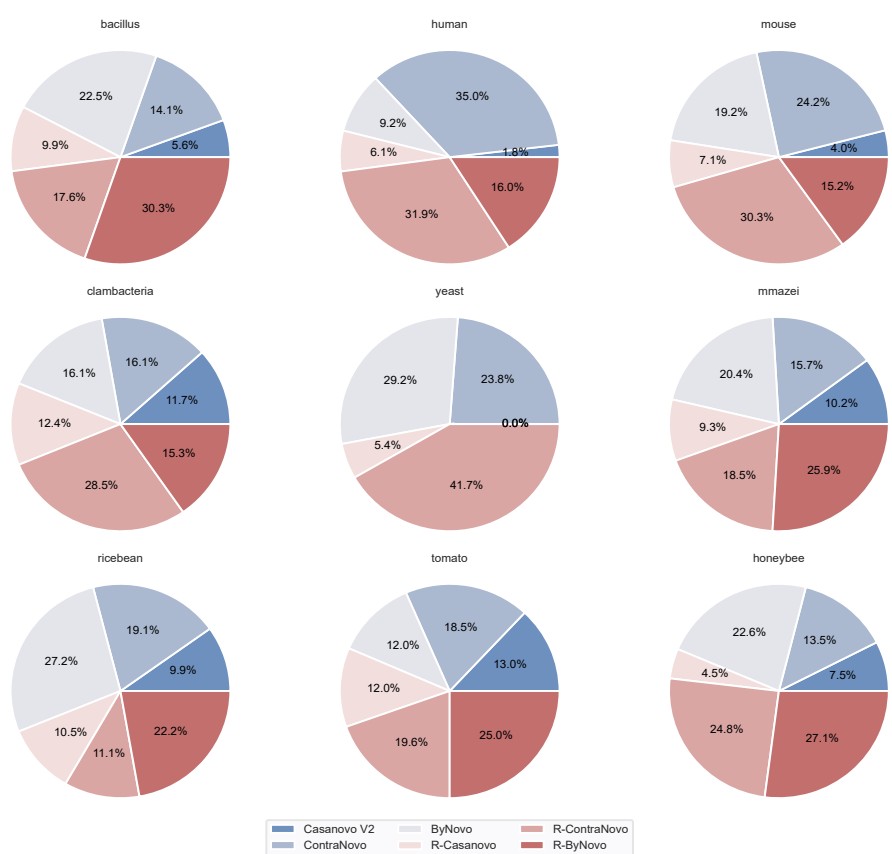

Figure 8: Unique-correctly selected percentage of base models. The numbers are calculated on 9-species-V1 dataset.

### A.5.5 COMPARISON WITH TRIVIAL ENSEMBLE METHODS

| | Methods | Bacillus | C. bacteria | Honeybee | Human | M.mazei | Mouse | Rice bean | Tomato | Yeast | Average |
|---|---|---|---|---|---|---|---|---|---|---|---|
| | HC | 0.830 | 0.702 | 0.771 | 0.781 | 0.804 | 0.799 | 0.819 | 0.811 | 0.742 | 0.784 |
| **AA** | MF | 0.813 | 0.685 | 0.748 | 0.737 | 0.783 | 0.782 | 0.788 | 0.792 | 0.750 | 0.764 |
| **Precision** | PM | 0.783 | 0.675 | 0.707 | 0.673 | 0.749 | 0.756 | 0.748 | 0.781 | 0.743 | 0.735 |
| | **RankNovo** | **0.874** | **0.746** | **0.81** | **0.802** | **0.84** | **0.828** | **0.859** | **0.844** | **0.816** | **0.824** |
| | HC | 0.705 | 0.508 | 0.598 | 0.628 | 0.643 | 0.575 | 0.706 | 0.682 | 0.621 | 0.630 |
| **Peptide** | MF | 0.709 | 0.513 | 0.598 | 0.611 | 0.645 | 0.564 | 0.702 | 0.677 | 0.666 | 0.632 |
| **Recall** | PM | 0.608 | 0.428 | 0.478 | 0.428 | 0.546 | 0.472 | 0.573 | 0.607 | 0.585 | 0.525 |
| | **RankNovo** | **0.738** | **0.539** | **0.63** | **0.642** | **0.672** | **0.583** | **0.733** | **0.691** | **0.703** | **0.660** |

Table 12: Performance comparison of RankNovo with three other trivial ensembling methods on 9-species-V1 test set. HC stands for selecting the candidate with the highest confidence score. MF stands for selecting the candidate most frequently predicted. PM stands for selecting the candidate closest to the precursor mass.

### A.5.6 MORE INFORMATION ABOUT TRAINING TIME AND INFERENCE TIME

| Model | Parameters (M) | Training Time (Day) | Infer. Cost (s/spectra) | Infer. Speed (spectras/s) |
|---|---|---|---|---|
| Casa. & R-Casa. | 47.3 | 3 | 0.127 | 7.87 |
| Contra. & R-Contra. | 68.6 | 4 | 0.173 | 5.78 |
| By. & R-By. | 49.7 | 3 | 0.169 | 5.92 |
| **RankNovo** | **50.5** | **4** | **/** | **/** |

Table 13: Summary of model parameters, training time, and inference speed of six base models and RankNovo.

| N. Infer | Candidates Collection (s/spectra) | Reranking (s/spectra) | Total Cost (s/spectra) | Infer. Speed (spectra/s) |
|---|---|---|---|---|
| 2 | 0.254 | 0.004 | 0.258 | 3.88 |
| 3 | 0.423 | 0.006 | 0.429 | 2.33 |
| 4 | 0.596 | 0.008 | 0.604 | 1.66 |
| 5 | 0.769 | 0.010 | 0.779 | 1.28 |
| 6 | 0.938 | 0.011 | 0.949 | 1.05 |

Table 14: RankNovo's inference speed when using different number of base models. The combination of base models refers to Table 6.

### A.5.7 ADDITIONAL RESULTS FOR THE REVERSE COMBINATION OF BASE MODELS

| Model Num. | Base Model Set |
|---|---|
| 2 | ByNovo, ContraNovo |
| 3 | ByNovo, ContraNovo, R-ContraNovo |
| 4 | ByNovo, ContraNovo, R-ContraNovo, R-ByNovo |
| 5 | ByNovo, ContraNovo, R-ContraNovo, R-ByNovo, R-Casanovo |
| 6 | ByNovo, ContraNovo, R-ContraNovo, R-ByNovo, R-Casanovo, Casanovo-V2 |

Table 15: Description of different combination of base models. The combinations are generated by sequentially removing the weakest model.

| | N. Train | N. Infer | Bacillus | C. bacteria | Honeybee | Human | M.mazei | Mouse | Rice bean | Tomato | Yeast | Average |
|---|---|---|---|---|---|---|---|---|---|---|---|---|
| **Amino Acid Precision** | 2 | 2 | 0.855 | 0.721 | 0.787 | 0.781 | 0.819 | 0.809 | 0.832 | 0.825 | 0.799 | 0.803 |
| | 3 | 3 | 0.864 | 0.733 | 0.799 | 0.798 | 0.832 | 0.824 | 0.844 | 0.836 | 0.812 | 0.816 |
| | 4 | 4 | 0.868 | 0.737 | 0.806 | 0.8 | 0.836 | 0.825 | 0.85 | 0.84 | 0.812 | 0.819 |
| | 5 | 5 | 0.871 | 0.743 | 0.808 | 0.804 | 0.838 | 0.829 | 0.855 | 0.842 | 0.818 | 0.823 |
| | 6† | 6 | **0.874** | **0.746** | **0.81** | **0.802** | **0.84** | **0.828** | **0.859** | **0.844** | **0.816** | **0.824** |
| **Peptide Recall** | 2 | 2 | 0.707 | 0.501 | 0.599 | 0.617 | 0.644 | 0.563 | 0.703 | 0.667 | 0.684 | 0.632 |
| | 3 | 3 | 0.719 | 0.518 | 0.613 | 0.638 | 0.653 | 0.577 | 0.709 | 0.677 | 0.694 | 0.644 |
| | 4 | 4 | 0.727 | 0.522 | 0.619 | 0.634 | 0.658 | 0.576 | 0.715 | 0.681 | 0.696 | 0.648 |
| | 5 | 5 | 0.732 | 0.530 | 0.624 | 0.638 | 0.663 | 0.582 | 0.727 | 0.685 | 0.702 | 0.654 |
| | 6† | 6 | **0.738** | **0.539** | **0.63** | **0.642** | **0.672** | **0.583** | **0.733** | **0.691** | **0.703** | **0.660** |

Table 16: Peptide recall evaluation of RankNovo on 9-species-V1 test set when the training base model set and the inference base model set are the same and vary. The symbol "†" indicates that the model is the final RankNovo mentioned in the main text. The model subsets here are created by sequentially removing the weakest model, as introduced in Table 15.

| | N. Train | N. Infer | Bacillus | C. bacteria | Honeybee | Human | M.mazei | Mouse | Rice bean | Tomato | Yeast | Average |
|---|---|---|---|---|---|---|---|---|---|---|---|---|
| **Amino Acid Precision** | 2 | 6 | 0.871 | 0.745 | 0.808 | 0.801 | 0.839 | 0.828 | 0.856 | 0.843 | 0.814 | 0.823 |
| | 3 | 6 | 0.871 | 0.745 | 0.809 | 0.802 | 0.839 | 0.829 | 0.856 | 0.844 | 0.815 | 0.823 |
| | 4 | 6 | 0.873 | 0.745 | 0.810 | 0.802 | 0.841 | 0.829 | 0.857 | 0.845 | 0.817 | 0.824 |
| | 5 | 6 | 0.872 | 0.743 | 0.81 | 0.802 | 0.839 | 0.829 | 0.858 | 0.843 | 0.817 | 0.824 |
| | 6† | 6 | **0.874** | **0.746** | **0.81** | **0.802** | **0.84** | **0.828** | **0.859** | **0.844** | **0.816** | **0.824** |
| **Peptide Recall** | 2 | 6 | 0.727 | 0.526 | 0.615 | 0.627 | 0.665 | 0.581 | 0.726 | 0.685 | 0.687 | 0.649 |
| | 3 | 6 | 0.732 | 0.529 | 0.623 | 0.632 | 0.665 | 0.583 | 0.727 | 0.685 | 0.698 | 0.653 |
| | 4 | 6 | 0.735 | 0.535 | 0.625 | 0.639 | 0.669 | 0.583 | 0.728 | 0.688 | 0.700 | 0.656 |
| | 5 | 6 | 0.738 | 0.533 | 0.628 | 0.639 | 0.673 | 0.584 | 0.735 | 0.690 | 0.701 | 0.658 |
| | 6† | 6 | **0.738** | **0.539** | **0.63** | **0.642** | **0.672** | **0.583** | **0.733** | **0.691** | **0.703** | **0.660** |

Table 17: Peptide recall evaluation of RankNovo on 9-species-V1 test set when the training base model set varies and the inference base model set is fixed. The symbol "†" indicates that the model is the final RankNovo mentioned in the main text. The model subsets here are created by sequentially removing the weakest model, as introduced in Table 15.

