# OpenReview forum: "RankNovo: A Universal Reranking Approach for Robust De Novo Peptide Sequencing"
_ICLR.cc/2025/Conference — Submitted to ICLR 2025_

### Official Review · Reviewer_GLyc · 2024-11-02

**Soundness:** 3
**Presentation:** 3
**Contribution:** 2
**Rating:** 5
**Confidence:** 3

**Summary:**

The paper presents the first deep reranking framework, RankNovo, based on a list-wise reranking approach, which enhances de novo peptide sequencing by leveraging the complementary strengths of multiple sequencing models. Experimental results show that RankNovo achieves state-of-the-art performance on de novo sequencing benchmarks, outperforming each of its component base models.  Moreover, RankNovo exhibits strong zero-shot generalization to unseen models that highlight its robustness and potential.

**Strengths:**

1. The paper introduces a deep learning-based reranking framework for peptide de novo sequencing, and focus on addressing the preferential bias challenges inherent in peptide sequencing.
2. The paper introduces two novel metrics, PMD and RMD, for accurate measurement of mass differences between peptides, which enable the model to more accurately distinguish complex similar sequences.
3. In the experiments, RankNovo surpasses each of its individual ensemble components on the 9-species-V1 and the 9-Species-v2 dataset, demonstrating superior performance in amino acid and peptide.
4. RankNovo generalizes effectively to unseen models in a zero-shot setting, which underscores the potential and value for future applications in de novo peptide sequencing.

**Weaknesses:**

1. The proposed RankNovo relies on the setting of multiple base models, which increases the computational cost.
2. In formula 10, L_coarse and L_fine are not clearly defined. It seems more appropriate to use L_PMD and L_RMD?
3. In formula (1) and formula (10), λ is confused.
4. The setting of λ in Equation 10 is unclear. Is the λ used in different tasks inconsistent? The author needs to clarify the setting of λ and the impact of λ on performance.
5. In order to study the influence of the number of base models and each model, you select five subsets. Could you explain the basis for sequentially removing the strongest model? Why not remove the poorest ones?
6. In Table 8, compared with the 5 models, the 6 models introduced the strongest ByNovo, but the average performance was reduced. What is the possible reason?
7. In Conclusion, there is relatively little content on the prospects and challenges of future optimization and applications, research limitations. It is suggested to supplement some to make it more comprehensive and forward-looking.

Minor issues:
1. typo error: Please confirm if the term ‘expriment’ is used correctly in the titles of Section 4 and 4.1 of the article.
2. There are three models in "The latter four models, ByNovo, R-ContraNovo, and R-ByNovo".

**Questions:**

Please see the questions and suggestions raised in the section Weaknesses above.

---

> ### Author Response · Authors · 2024-11-20
> **To Reviewer GLyc**
>
> We thank you for your reviews and address your concerns as follows.
>
> **Q1**: The proposed RankNovo relies on the setting of multiple base models, which increases the computational cost.
>
> A1: Your assessment is reasonable. We have now incorporated comprehensive details of the inference speeds for RankNovo in the new section A.5.6. As anticipated, RankNovo's inference speed is indeed slower than single-model approaches. However, the reranking process itself is not the primary time constraint. The majority of RankNovo's inference time is consumed in gathering peptide candidates from base models, as these require sequential autoregression and beam search decoding, while RankNovo's inference involves only a single attention forward pass.
>
> As we scale from 2 to 6 base models in RankNovo, the inference time increases approximately linearly, with inference speed decreasing proportionally. This increased computational cost is an inherent characteristic of reranking frameworks and represents an unavoidable trade-off compared to single-model approaches. However, RankNovo effectively leverages this additional inference time to achieve superior performance levels unattainable by single models. This inference time-performance trade-off can be flexibly adjusted by modifying the number of candidates.
>
> In the context of de novo peptide sequencing, RankNovo's significance lies in introducing a novel approach that allows researchers to optionally scale up inference time in exchange for enhanced performance. This represents the first such option in the field.
>
> **Q2**: In formula 10, L_coarse and L_fine are not clearly defined. It seems more appropriate to use L_PMD and L_RMD?
>
> A2: We agree with the your suggestion. To enhance clarity and consistency, we have replaced the terms L_coarse and L_fine with L_PMD and L_RMD in Formula 10. These new terms more accurately reflect the nature of the loss functions, aligning with the definitions provided earlier in the paper.
>
> **Q3**: In formula (1) and formula (10), λ is confused.
>
> A3:  We're sorry for the confusion caused by the misuse of notations. To resolve this confusion, we have now changed all λs representing mass-to-charge ratio to μ, while λ is only used in formula (10) representing the weight of aggregating PMD objective and RMD objective.
>
> **Q4**: Is the λ used in different tasks inconsistent? The author needs to clarify the setting of λ and the impact of λ on performance.
>
> A4: Thank you for pointing this out. We apologize for the oversight in clarifying this aspect. To address this, we have now explicitly stated in line 348 of the revised manuscript that **λ** is consistently set to **0.5** across all tasks. The parameter **λ** is used to aggregate the two training objectives, PMD (Prefix Mass Difference) and RMD (Residue Mass Difference). To analyze the impact of **λ** on performance, we conducted an ablation study to evaluate the effects of using PMD alone (**λ=1**) or RMD alone (**λ=0**) as the training objective. The results of this study are provided in **Table 4** and discussed in detail in **Appendix A.4.2**. Our findings show that combining PMD and RMD (**λ=0.5**) yields superior results compared to using either PMD or RMD individually. This demonstrates that a balanced integration of both objectives is crucial for achieving optimal performance.

---

> > ### Author Response · Authors · 2024-11-20
> > **To Reviewer GLyc Part Ⅱ**
> >
> > **Q5**: In order to study the influence of the number of base models and each model, you select five subsets. Could you explain the basis for sequentially removing the strongest model? Why not remove the poorest ones?
> >
> > A5: RankNovo can utilize up to six base models, forming 56 different subsets containing two to five base models. Evaluating all possible combinations would be computationally prohibitive and could lead to noisy and complex results. Thus, it was necessary to select representative subsets to study the influence of the number of base models used during training and inference. In our original experimental design, we created five subsets by sequentially removing the strongest model for two primary reasons:
> > 1. **Comprehensive Experimental Insights**: This configuration allowed us to systematically assess performance improvements as the model set size increased from 2 to 6. During this process, **R-ByNovo**, **R-ContraNovo**, **ContraNovo**, and **ByNovo** were sequentially added during training. Observing consistent performance improvements with increasing model set size demonstrated that all four models positively contribute to RankNovo’s training. Furthermore, comparing RankNovo trained with two base models to the individual performance of **Casanovo-V2** or **R-Casanovo** highlighted the contributions of these two relatively weaker models.
> > 2. **Pronounced Performance Differences**: Sequentially removing the strongest model was hypothesized to yield more pronounced performance differences between configurations, making overall trends more apparent. Your suggestion to remove the poorest-performing models is indeed valid. To investigate whether this alternative approach impacts the observed trends, we conducted additional experiments using your proposed configuration, with detailed results presented in **Appendix A.5.7**.
> >
> > Your suggestion to remove the poorest-performing models is indeed valid. To investigate whether this alternative approach impacts the observed trends, we conducted additional experiments using your proposed configuration, with detailed results presented in **Appendix A.5.7**.
> >
> > **Findings**:
> > - The overall trend remains consistent under this alternative setting. Knowledge acquired by certain base models can still be adapted to other base models in a zero-shot manner during inference. Increasing the number of base models during training continues to yield better overall performance.
> >  - However, as expected, the performance trends are less pronounced compared to our original configuration, where the strongest models were sequentially removed. This additional analysis strengthens our understanding of the interplay between base model selection and RankNovo’s overall performance.
> >
> > **Q6**: In Table 8, compared with the 5 models, the 6 models introduced the strongest ByNovo, but the average performance was reduced. What is the possible reason?
> >
> > A6: The primary performance metric for peptide *de novo* sequencing is **peptide recall**, not amino acid precision [1]. This is because, in proteomics experiments, a tandem mass spectrum (MS/MS) identification is considered successful only when all residues in the peptide are correctly sequenced. While amino acid-level metrics provide additional insights—particularly for earlier *de novo* sequencing algorithms evaluated on smaller datasets [2][3]—they are secondary to peptide recall. From this perspective, RankNovo demonstrates consistent improvement with the inclusion of additional base models during training, as evidenced by the monotonic increase in **peptide recall** from 0.647 to 0.660. The relationship between peptide recall and amino acid precision is not necessarily directly proportional. The slight decline in amino acid precision observed after incorporating ByNovo can be attributed to RankNovo's enhanced ability to identify shorter peptides, which increases peptide recall. However, this improvement may come at the expense of selecting less reliable candidates for certain longer peptides. To illustrate, consider a mini case observed in the Bacillus species dataset (V1). For three MS/MS spectra with the following true sequences:
> > - `KEYAVVNIDK`
> > - `IVFPEGIDER`
> > - `NTPGVTGFVGSAGSGSKPTPIIPGEAETIIKR`
> >
> > The five-model version predicted:
> > - `EYKAVVNLDK`
> > - `LVFPVAEDER`
> > - `NTPGVTGFVGSAGSGSKPTPLLPGEAETLLKR`
> >
> > In contrast, the six-model version predicted:
> > - `KEYAVVNIDK`
> > - `IVFPEGIDER`
> > - `N+0.984TPGVTGGATYAPTSKPTPLLPGEAETLLKR`
> >
> > This resulted in the following metrics:
> > - **Five-model version**: Peptide recall = 0.333, Amino acid precision = 0.884
> > - **Six-model version**: Peptide recall = 0.667, Amino acid precision = 0.846
> >
> > This case demonstrates that an increase in peptide recall can coincide with a slight decrease in amino acid precision, which is a common tradeoff in such scenarios. Ultimately, the improvement in peptide recall highlights the enhanced performance of RankNovo in achieving its primary objective.

---

> > > ### Author Response · Authors · 2024-11-20
> > > **To Reviewer GLyc Part Ⅲ**
> > >
> > > **Q7**: In Conclusion, there is relatively little content on the prospects and challenges of future optimization and applications, research limitations. It is suggested to supplement some to make it more comprehensive and forward-looking.
> > >
> > > A7: Your suggestion is very pertinent. In response, we have revised the Conclusion section to address the current limitations, future development directions, and potential impact on the de novo sequencing field. Specifically, we acknowledge that the primary limitation is the relatively slow inference speed. Future research could investigate efficient candidate sampling techniques, such as implementing base models with partially shared weights to reduce computational complexity. While RankNovo faces speed constraints, it represents a pioneering deep reranking framework that enables flexible balancing between inference time and performance, introducing an innovative approach to performance enhancement. We anticipate that, influenced by RankNovo, future algorithms in this field will benefit from the synergistic approach of simultaneously improving single-model performance and developing advanced reranking strategies.
> > >
> > > [1] NovoBench: Benchmarking Deep Learning-based De Novo Peptide Sequencing Methods in Proteomics
> > >
> > > [2] PEAKS: powerful software for peptide de novo sequencing by tandem mass spectrometry
> > >
> > > [3] PepNovo:  De Novo Peptide Sequencing via Probabilistic Network Modeling

---

### Official Review · Reviewer_2GzM · 2024-11-03

**Soundness:** 3
**Presentation:** 3
**Contribution:** 2
**Rating:** 5
**Confidence:** 4

**Summary:**

The paper presents RankNovo, a list-wise deep reranking framework for de novo peptide sequencing that uses outputs from multiple base models, applying axial attention and novel metrics for effective peptide reranking.

**Strengths:**

1. Introduces a unique list-wise reranking approach that effectively leverages outputs from multiple de novo peptide sequencing models.
2. Introduces PMD and RMD metrics, designed for precise quantification of mass differences between peptides.
3. The model demonstrates significant improvements over existing methods.

**Weaknesses:**

1. The approach heavily depends on outputs from existing peptide sequencing models, which may limit its novelty.
2. The conclusion is somewhat simple and could be expanded to discuss the limitations of the current approach and potential future research directions.

**Questions:**

1. RankNovo incorporates six de novo sequencing models. Is there any rationale for the selection of these?
2. See weakness.

---

> ### Author Response · Authors · 2024-11-20
> **To Reviewer 2GzM**
>
> We thank you for your reviews and address your concerns as follows.
>
> **Q1**: The approach heavily depends on outputs from existing peptide sequencing models, which may limit its novelty.
>
> A1: Reranking is a well-established technique across numerous natural language processing tasks, including question answering [1][2], recommendation systems [3], and recent large language model answer selection [4]. This technique involves collecting various candidates for a single query and selecting the optimal one among them. In the context of de novo sequencing, while reranking does utilize outputs from different existing models, this should not be viewed as a weakness or limitation. On the contrary, the reranking framework demonstrates excellent flexibility - regardless of base models and the source of candidates, RankNovo consistently delivers performance improvements (see Fig 3 (A) and Appendix A.5.1). RankNovo has illuminated an alternative pathway for enhancing sequencing task performance, distinct from the traditional approach of improving individual models. We anticipate that, influenced by RankNovo, future algorithmic development in this field will benefit from the synergistic interaction between two approaches: "improving single model performance" and "developing superior ranking strategies." This dual-pathway advancement represents RankNovo's fundamental contribution to the field.
>
> **Q2**: The conclusion is somewhat simple and could be expanded to discuss the limitations of the current approach and potential future research directions.
>
> A2: Your suggestion is very pertinent. In response, we have revised the Conclusion section to address the current limitations, future development directions, and potential impact on the de novo sequencing field. Specifically, we acknowledge that the primary limitation is the relatively slow inference speed. Future research could investigate efficient candidate sampling techniques, such as implementing base models with partially shared weights to reduce computational complexity. While RankNovo faces speed constraints, it represents a pioneering deep reranking framework that enables flexible balancing between inference time and performance, introducing an innovative approach to performance enhancement. We anticipate that, influenced by RankNovo, future algorithms in this field will benefit from the synergistic approach of simultaneously improving single-model performance and developing advanced reranking strategies.

---

> > ### Author Response · Authors · 2024-11-20
> > **To Reviewer 2GzM Part Ⅱ**
> >
> > **Q3**: Is there any rationale for the selection of the six base models of RankNovo?
> >
> > A3: Yes, the criteria for selecting the six base models are described in detail in **Appendix A.2.3** and can be summarized as follows:
> > 1. **Avoiding Data Leakage**: The training datasets for the base models must be carefully controlled to ensure there is no data leakage, preserving the integrity of the evaluation.
> > 2. **Diversity of Algorithms**: The base models should employ different underlying algorithms to capture varying data preferences, thereby enriching the ensemble’s capacity to rerank effectively.
> > 3. **Performance Proximity**: The selected base models should exhibit comparable performance levels to ensure that no single model dominates the ensemble.
> > 4. **Optimal Performance**: The performance of the base models should be as strong as possible to maximize their contributions to the reranking process.
> >
> > Based on these criteria:
> > - **Criterion (1) and (4)** restrict the selection to models trained on **Massive-KB** and evaluated on **Nine-species-V1/2**, as **Massive-KB** is the largest available training dataset, resulting in the best-performing models. Consequently, de novo sequencing algorithms such as GraphNovo [5] and Π-HelixNovo [6], which do not meet this criterion, were excluded.
> > - **Criterion (4)** necessitates including the previous state-of-the-art model, ContraNovo [7].
> > - **Criterion (3)** excludes weaker models, such as those performing below Casanovo-V2 [8], as these models would likely fail to contribute to reranking. As shown in **Fig. 3(B)** and **Appendix Fig. 8**, even Casanovo-V2 uniquely solves less than 10% of the spectra, making weaker models ineffective for this task.
> >
> > After this filtering process, only **ContraNovo** and **Casanovo-V2** remained among publicly available models. To satisfy **Criterion (2)** for algorithmic diversity, we trained four additional models—**ByNovo, R-Casanovo, R-ContraNovo, and R-ByNovo**—to complement the ensemble.
> >
> > This selection strategy ensures a diverse and high-performing base for RankNovo.
> >
> > [1] RankT5: Fine-Tuning T5 for Text Ranking with Ranking Losses
> >
> > [2] RankQA: Neural Question Answering with Answer Re-Ranking
> >
> > [3] Personalized re-ranking for recommendation
> >
> > [4] Aggregation of Reasoning: A Hierarchical Framework for Enhancing Answer Selection in Large Language Models
> >
> > [5] Mitigating the missing-fragmentation problem in de novo peptide sequencing with a two-stage graph-based deep learning model
> >
> > [6] Introducing π-HelixNovo for practical large-scale de novo peptide sequencing
> >
> > [7] ContraNovo: A Contrastive Learning Approach to Enhance De Novo Peptide Sequencing
> >
> > [8] Sequence-to-sequence translation from mass spectra to peptides with a transformer model

---

### Official Review · Reviewer_irm2 · 2024-11-03

**Soundness:** 3
**Presentation:** 3
**Contribution:** 3
**Rating:** 6
**Confidence:** 4

**Summary:**

This paper presents the first deep learning-based reranking framework that enhances de novo peptide sequencing by leveraging the complementary strengths of multiple sequencing models. RankNovo scores the output sequences of multiple de novo models to train the model (training phase) or to obtain the optimal sequence through scoring (inference phase). The paper also introduces novel metrics such as PMD and RMD.

**Strengths:**

1. The paper demonstrates a high degree of novelty by proposing a method that utilizes predicted sequences from multiple de novo models, challenging the traditional single-model paradigm.
2. It achieves performance that surpasses all baselines and base models on two datasets.
3. The introduction of novel metrics such as PMD and RMD is likely the first of its kind in de novo sequencing, providing valuable reference points.

**Weaknesses:**

1. The performance improvement seems modest. RankNovo's peptide recall only surpasses the baseline by 0.037 (V1) and 0.023 (V2). Since RankNovo is trained and inferred based on these base models, it theoretically can only outperform them. The evaluation of RankNovo's performance should be based on its improvement relative to SOTA baselines. Otherwise, for such minor improvements, why not simply use the predictions from other models instead of inference multiple models through RankNovo?

2. There seems to be a lack of analysis regarding the parameter number, training time, and inference time of RankNovo compared to other de novo models, which should be addressed in the experimental section. Given that RankNovo requires inference results from multiple (2 to 6) other models during training and inference, it can be inferred that its training and inference times would be slower than those of other models.

**Questions:**

1. The definition of robustness in the paper seems somewhat ambiguous. On one hand, it appears to refer to the model's robustness to noise in mass spectrometry data (Lines 12–14, Lines 81–84), but RankNovo might not sufficient to address this issue. On the other hand, it seems to refer to the issue of having inconsistent baseline models available during training and testing phases.


2. Based on the disadvantage mentioned earlier regarding modest performance improvement, there should be experiments to demonstrate whether naive methods (instead of RankNovo) can also achieve reasonable results. For instance, during inference, for multiple predicted peptides from different base models, one could select: (1) the peptide with the highest score from a model output, (2) the most frequently predicted peptide, or (3) the peptide closest to the precursor mass, and compare the performance of these methods against RankNovo. This approach would help demonstrate that RankNovo indeed learns some knowledge from the predicted results of multiple models.

---

> ### Author Response · Authors · 2024-11-20
> **To Reviewer irm2**
>
> Thank you for your detailed comments. We re-list and address your concerns as follows. The order of these concerns are rearranged to better express our opinions.
>
> **Q1**: RankNovo should be evaluated against SOTA baselines, or others may favor a stronger single model instead.
>
> A1: ContraNovo [1], which is referenced in the initial three lines of our manuscript, represents the current published state-of-the-art baseline for Nine-species-V1 and Nine-species-V2 datasets. In our research, we incorporated ContraNovo as one of six base models in our reranking framework, and Tables 1 and 2 present comprehensive comparative analyses between RankNovo and ContraNovo. Therefore, RankNovo has already been evaluated against SOTA baselines, and shows superior performance both on Nine-species-V1 and Nine-species-V2.
>
> Our reranking framework includes not only published models but also four internally developed base models: ByNovo, R-Casanovo, R-ContraNovo, and R-ByNovo. These models demonstrate performance comparable to or exceeding that of ContraNovo, with ByNovo being a notable example of superior performance. Tables 1 and 2 clearly demonstrate RankNovo's performance advantages over all baseline models.
>
> Therefore, we believe that empirical evidence substantiates RankNovo's superior performance relative to state-of-the-art baselines.
>
> **Q2**: RankNovo's peptide recall improvement compared to base models are modest, which is expected since RankNovo uses ensemble and reranking framework.
>
> A2: We acknowledge your observation regarding the scale of RankNovo's improvements, but we would like to contextualize these results within recent advances in the field.
> For perspective, the state-of-the-art ContraNovo [1] demonstrated peptide recall improvements of 0.051 and 0.038 over its predecessor, Casanovo-V2 [2]. Subsequently, a recent preprint work PrimeNovo [3] achieved gains of 0.020 and 0.025 over ContraNovo. In this context, RankNovo's improvements of 0.042 (Version 1) and 0.029 (Version 2) over ContraNovo, and 0.037 (Version 1) and 0.023 (Version 2) over the strongest base model ByNovo (also introduced in this work) represent significant advancements in the field. These incremental improvements are particularly meaningful in practical proteomics applications, where each additional correct peptide identification contributes to experimental accuracy. Furthermore, RankNovo's significance extends beyond metric improvements alone. The framework introduces a novel reranking methodology that complements existing sequencing models, establishing an alternative approach to performance enhancement that diverges from conventional single-model optimization strategies. We anticipate that this dual-approach paradigm will influence future algorithmic developments in the field, fostering innovation through the integration of both methodologies.
>
> **Q3**: RankNovo should be slower than other single model frameworks, and the parameter number, training time, and inference time should be provided.
>
> A3: We have incorporated comprehensive details regarding parameter counts, training time, and inference speeds for our six base models and RankNovo in the new section A.5.6. In summary, RankNovo comprises 50.5M parameters and requires 4 days of training utilizing four 40GB A100 GPUs, which is comparable to the base models. As anticipated, RankNovo's inference speed is indeed slower than single-model approaches. However, the reranking process itself is not the primary time constraint. The majority of RankNovo's inference time is consumed in gathering peptide candidates from base models, as these require sequential autoregression and beam search decoding, while RankNovo's inference involves only a single attention forward pass.
>
> As we scale from 2 to 6 base models in RankNovo, the inference time increases approximately linearly, with inference speed decreasing proportionally. This increased computational cost is an inherent characteristic of reranking frameworks and represents an unavoidable trade-off compared to single-model approaches. However, RankNovo effectively leverages this additional inference time to achieve superior performance levels unattainable by single models. This inference time-performance trade-off can be flexibly adjusted by modifying the number of candidates.
>
> In the context of de novo peptide sequencing, RankNovo's significance lies in introducing a novel approach that allows researchers to optionally scale up inference time in exchange for enhanced performance. This represents the first such option in the field.

---

> > ### Author Response · Authors · 2024-11-20
> > **To Reviewer irm2 Part Ⅱ**
> >
> > **Q4**: The definition of robustness in the paper seems to refer to the model's robustness to noise in mass spectrometry data, but RankNovo might not sufficient to address this issue. On the other hand, it seems to refer to the issue of having inconsistent base models available during training and testing phases.
> >
> > A4: We appreciate your insightful observation and acknowledge the confusion arising from our use of the term "robustness." Our intended meaning refers to RankNovo's capability to select superior candidate peptides from multiple base models' predictions, even when certain base models were not incorporated during training. To avoid confusion with whether a model will be affected by the noises in mass spectrum data, we have now corrected the misuse of "robustness" and replaced them with other words in the revised manuscripts.
> >
> > Regarding your observation about RankNovo's potential limitations in addressing mass spectrum data noise heterogeneity, we posit that such heterogeneity is not a problem to be solved, but rather an inherent and essential feature of de novo sequencing. One of RankNovo's contributions is highlighting this fundamental feature to researchers in the field, and our reranking framework attempts to leverage this feature constructively. While our approach may have limitations, we believe it provides valuable insights for future research directions in this domain.
> >
> > **Q5**: Whether some naive methods can also achieve reasonable results, such as (1) the peptide with the highest score from a model output (2) the most frequently predicted peptide (3) the peptide closest to the precursor mass.
> >
> > A5: We appreciate this insightful question. We have evaluated these three methods and included a performance comparison with RankNovo in Appendix A.5.5 of our revised manuscript. Our findings show that methods (1) and (2) demonstrate modest improvements in peptide recall compared to the best ByNovo baseline on Nine-species-V1, with increases of 0.008 and 0.010 respectively. However, these improvements are substantially lower than RankNovo's 0.037 gain. Method (3) resulted in a significant decrease in peptide recall from 0.623 to 0.525. This decline can be attributed to the fact that most candidate peptides already closely match the precursor mass. Method (3)'s exclusive focus on mass difference, while disregarding other semantic information, proves counterproductive. Methods (1) and (2) show some improvement by leveraging collective intelligence, as expected. However, these approaches do not utilize semantic information, spectral data, and candidate peptide features as comprehensively as RankNovo's data mining and machine learning methodology. Therefore, to achieve optimal performance, deep learning reranking approaches as exemplified by RankNovo  remain essential.
> >
> > [1] Contranovo: A contrastive learning approach to enhance de novo peptide sequencing
> >
> > [2] Sequence-to-sequence translation from mass spectra to peptides with a transformer model
> >
> > [3] π-PrimeNovo: An Accurate and Efficient Non-Autoregressive Deep Learning Model for De Novo Peptide Sequencing

---

> > > ### Comment · Reviewer_irm2 · 2024-11-26
> > >
> > > Thank you very much for your response. The authors have effectively addressed all of my questions, and I will maintain my positive score.

---

### Official Review · Reviewer_pMKc · 2024-11-03

**Soundness:** 3
**Presentation:** 3
**Contribution:** 4
**Rating:** 6
**Confidence:** 4

**Summary:**

It is an interesting paper that proposes a deep learning-based reranking framework and introduces two new metrics, PMD and RMD, to characterize quality differences at the peptide and residue levels. The new framework leverages the strengths of multiple de novo peptide sequencing models to achieve improved performance through reranking.

**Strengths:**

- The paper is written very clearly and is easy to understand.
- The motivation is highly meaningful, revealing that different models have distinct advantages.
- A general framework is proposed, adaptable to any de novo peptide sequencing model.
- The experiments are thorough.

**Weaknesses:**

- The discussion of transformer-based approaches in the introduction is insufficient.
- The meaning of some mathematical symbols is unclear.

**Questions:**

- Recently, many transformer-based de novo peptide sequencing methods have emerged, such as HelixNovo[1], InstaNovo[2], AdaNovo[3], PrimeNovo[4], GraphNovo[5], etc. The discussion of these transformer-based approaches in the introduction is insufficient.

- In line 270, M  represents the residue mass, but the same symbol is also used in line 306 to denote prefix mass. It would be better to add a subscript or identifier to distinguish prefix mass or, alternatively, use a different symbol to represent it.

- In the mathematical formula (7) on line 309, should  \overline{m}_{q\tilde{j}} - \overline{m}_{k\tilde{j}} be corrected to  \overline{m}_{qi} - \overline{m}_{k\tilde{j}}  to represent the prefix mass difference more accurately?

[1] π-HelixNovo for practical large-scale de novo peptide sequencing

[2] De novo peptide sequencing with InstaNovo: Accurate, database-free peptide identification for large scale proteomics experiments

[3] AdaNovo: Adaptive \emph{De Novo} Peptide Sequencing with Conditional Mutual Information

[4] π-PrimeNovo: An Accurate and Efficient Non-Autoregressive Deep Learning Model for De Novo Peptide Sequencing

[5] Mitigating the missing-fragmentation problem in de novo peptide sequencing with a two-stage graph-based deep learning model

---

> ### Author Response · Authors · 2024-11-20
> **To Reviewer pMKc**
>
> We thank you for your reviews and address your concerns as follows.
>
> **Q1**: The discussion of transformer-based approaches in the introduction is insufficient.
>
> A1: We have significantly enhanced the discussion of transformer-based approaches in both the Introduction and Related Work sections. The revised text provides a more comprehensive review of recent advancements in transformer architectures, with a particular focus on their relevance to de novo sequencing and ensemble methods. These additions aim to contextualize our work more effectively within the broader research landscape. The updated content can be found on lines 74–78 and 136–138 of the revised manuscript.
>
> **Q2**: In line 270, M represents the residue mass, but the same symbol is also used in line 306 to denote prefix mass. It would be better to add a subscript or identifier to distinguish prefix mass or, alternatively, use a different symbol to represent it.
>
> A2: We sincerely apologize for any confusion caused by the inconsistent notation. In the revised manuscript, we have clarified this issue by ensuring that 'M' consistently represents the residue mass throughout the paper. Additionally, we have added summation notation in line 306 to explicitly and accurately denote the prefix mass, thus eliminating potential ambiguity.
>
> **Q3**: In the mathematical formula (7) on line 309, should \overline{m}{q\tilde{j}} - \overline{m}{k\tilde{j}} be corrected to \overline{m}{qi} - \overline{m}{k\tilde{j}} to represent the prefix mass difference more accurately?
>
> A3: Thank you for catching this error. You are correct, and we have rectified this typographical mistake in formula (7) on line 309 of the revised manuscript. The correction now accurately reflects the intended representation of the prefix mass difference. We appreciate your careful review and attention to this detail.

---

### Author Response · Authors · 2024-11-20
**General Response**

We sincerely thank all reviewers for their thoughtful and constructive feedback. We have carefully addressed each comment and revised the manuscript accordingly. To ensure clarity, all revised sections are highlighted in blue. Below, we summarize the primary revisions made to the paper:

**Conclusion Section**: We have substantially revised the Conclusion to explicitly discuss the current limitations of our approach, outline promising directions for future work, and elaborate on the potential impact of our research on the field of de novo sequencing.
Introduction and Related Work: We have enriched the discussion of transformer-based approaches in both the Introduction and Related Work sections to better contextualize our contributions within the existing body of work.

**Performance Comparison**: To provide a more comprehensive evaluation, we have included a performance comparison between RankNovo and three naïve ensemble methods in **Appendix A.5.5**.

**Model Characteristics**: Additional information on model sizes, training costs, and inference speeds has been incorporated into **Appendix A.5.6**, offering deeper insights into the practicality and efficiency of our approach.

**Subset Analysis**: We now include detailed results for a series of base model subsets generated by sequentially removing the poorest-performing model, which are presented in **Appendix A.5.7**. This analysis further illustrates the robustness of our methodology.

**Typographical and Notation Corrections**: We have addressed typographical errors and clarified notation misuses throughout the manuscript to improve overall readability and precision.

For detailed responses and explanations, we refer to the official comments provided by the reviewers. We hope these revisions meet your expectations and enhance the quality and clarity of our work.

---

### Author Response · Authors · 2024-11-25
**Happy to address remaining questions**

We are grateful for the reviewers' thoughtful feedback.  We hope that our response has addressed all the issues raised by the reviewers, and that they would consider updating their scores accordingly. As we move towards the end of the public discussion phase, we welcome any additional questions or points requiring further clarification. Many thanks, The authors.

---

### Meta-Review · Area_Chair_cJmu · 2024-12-22

**Metareview:**

The paper considers the problem of de-novo peptide sequencing and introduces a deep reranking framework, RankNovo, to rerank the
candidate peptides from a set of base models using a list-wise strategy. The approach is demonstrated on de-novo sequencing benchmarks and outperforms each base models.

The paper is well written and the methodology sound and intuitive. The experimental study is extensive. In particular the analysis of the contribution of each base models is interesting. The AC and reviewers appreciate the additional discussion and empirical results provided during rebuttal (e.g. comparison with naive ensemble approaches). However, the novelty and significance of the contributions remain somewhat limited for ICLR. Indeed several reranking approaches have already been proposed for NLP, the approach yields modest performance improvements and the fact that it outperforms component base models is not surprising.

**Additional Comments On Reviewer Discussion:**

The points raised by the reviewers concerned confusing notation, the need for more discussion on transformer-based approaches, additional analysis on time and parameter complexity, selection of the based models, novelty and limited benefits, among others. The AC and reviewers appreciate the authors' clarifying points and additional experiments. However the marginal novelty and limited benefits remain and the AC believes that the paper is better suited for a computational biology conference.
As a remark, it would be interesting to study the influence of varying lambda to balance peptide-level and residual-level losses beyond the extreme cases of 0 and 1 reported in the rebuttal.

---

### Decision · Program_Chairs · 2025-01-22

Reject